# The emergence of sparse attention:
# impact of data distribution and benefits of repetition

**Nicolas Zucchet**[*]
ETH Zürich

**Francesco D'Angelo**
EPFL

**Andrew Lampinen**[†]
Google DeepMind

**Stephanie Chan**[†]
Google DeepMind

## Abstract

Emergence is a fascinating property of large language models and neural networks more broadly: as models scale and train for longer, they sometimes develop new abilities in sudden ways. Despite initial studies, we still lack a comprehensive understanding of how and when these abilities emerge. To address this gap, we study the emergence over training of sparse attention, a critical and frequently observed attention pattern in Transformers. By combining theoretical analysis of a toy model with empirical observations on small Transformers trained on a linear regression variant, we uncover the mechanics driving sparse attention emergence and reveal that emergence timing follows power laws based on task structure, architecture, and optimizer choice. We additionally find that repetition can greatly speed up emergence. Finally, we confirm these results on a well-studied in-context associative recall task. Our findings provide a simple, theoretically grounded framework for understanding how data distributions and model design influence the learning dynamics behind one form of emergence.

Scaling has been central to the recent success of large language models [1–5], with scaling laws [6, 7] describing how increased model size, data, and training consistently improve average performance. However, beneath this macroscopic predictability, model performance on specific tasks often reveals capabilities that appear suddenly beyond critical scaling thresholds – a phenomenon known as emergence [8–11]. While recent studies have begun to characterize how emergence can appear after a critical training time [12–18], a comprehensive scientific understanding remains elusive.

This work explores sparse attention as a lens to understand emergence during training. The formation of sparse attention – where Transformers' attention layers focus on a small subset of critical tokens – coincides with sudden performance improvements in many emergent behaviors, including in-context learning abilities [12, 15] and factual recall [18]. We investigate why the development of sparse attention can lead to abrupt performance improvements and reveal how characteristics of the training data influence the speed of emergence. We make two key contributions:

- We design a variant of linear regression that specifically requires Transformers to learn to focus on a few tokens within the context (Section 2). This analytically tractable task allows us to mathematically characterize, in a toy model, the mechanics behind the emergence of sparse attention and precisely quantify how shorter sequences and repetition accelerate emergence.

- We apply our sparse attention framework to explain the emergence of in-context learning in an associative recall task (Section 3). Our theoretical predictions successfully account for how data influences the emergence speed of the induction head that solves this task.

Overall, our results suggest that sparse attention may provide a unifying perspective for understanding seemingly diverse emergence phenomena in large language models. They additionally highlight the potential practical benefits of repetition for accelerating the formation of specific neural circuits.

---

[*]Correspondance to `nzucchet@ethz.ch`. [†]Advisory capacity only.

39th Conference on Neural Information Processing Systems (NeurIPS 2025).

# 1   Motivation

**Can emergence be predicted?** Emergence in large language models not only challenges our scientific understanding of how they acquire new skills but also poses AI safety issues [19] due to its unpredictable nature, while additionally complicating frontier model development. To address this unpredictability, researchers have proposed several progress metrics under which scaling has more predictable consequences, including validation loss [20], metrics that reward partial progress [21, 22], and those that employ mechanistic interpretability-motivated measures [23]. However, a significant limitation is that these metrics typically can only be derived post-emergence [24]. An alternative approach demonstrates that fine-tuned models' performance on certain tasks can predict whether the ability to solve the task will emerge in larger models [17]. Our work explores predictability from a different angle: understanding how the data distribution influences the learning time at which emergence occurs.

**Is there a link between emergence and sparse attention?** The driving hypothesis underlying this work is that learning of sparse attention patterns is particularly prone to produce sharp transitions in behavior during training – i.e., the sudden emergence of new capabilities. There is a statistical argument behind this intuition: when the target a Transformer has to predict depends only on a few tokens within the context, these tokens initially have very low weight in the prediction of the model, as attention typically starts uniform. Initial progress is therefore slow and the more targeted (thus sparse) attention is, the faster learning becomes. Multiple empirical observations strengthen our hypothesis. The induction head [12], whose formation coincides with the sudden emergence of certain in-context learning abilities, fundamentally relies on the combination of two attention layers with sparse patterns. Similarly, the mechanisms underlying factual recall in large language models [23, 25] demonstrate sparse attention properties and show emergent learning dynamics [18]. These specific emergent phenomena appear to be linked to the development of sparse attention mechanisms, suggesting the existence of a potential causal relationship between the two.

**The intriguing interplay between repetition and emergence.** While data diversity is generally considered a gold standard in machine learning, a growing body of evidence suggests that repetition can actually accelerate emergence over training. Chan et al. [13] demonstrated that showing some tasks more often favors the emergence of in-context learning. Charton and Kempe [16] revealed that repeating a subset of examples more frequently than others dramatically accelerates Transformers' ability to solve certain arithmetic tasks. This pattern extends beyond mathematical reasoning, as Zucchet et al. [18] (and to some extent Allen-Zhu and Li [26]) showed that repeating biographies of specific individuals speeds up the development of circuits critical for factual recall from model parameters. Collectively, these results establish repetition as a fundamental property of data distributions that can systematically influence emergence timing, thus justifying integrating it in our model to better understand its role.

# 2   Theoretical insights on an attention-based linear regression task

To illustrate how emergence can arise from sparse attention learning, we introduce a variant of linear regression that additionally requires selecting a relevant token from the input sequence. We introduce this task alongside a minimal attention-based toy model for solving it (Section 2.1), theoretically analyze its learning dynamics both without (Section 2.2) and with repetition (Section 2.3), and empirically show that our findings extend to more realistic Transformers (Section 2.4).

## 2.1   The single-location linear regression task

We consider the following supervised learning task. We are given an input sequence $(x_t)_{t=1}^{T}$ of length $T$ in which each token $x_t \in \mathbb{R}^d$ is drawn i.i.d. from a zero-mean normal distribution with variance $1/d$ and aim to predict the target $y^*$ given by

$$y^* = W^* x_T \tag{1}$$

Here, the target weight matrix $W^* \in \mathbb{R}^{d \times d}$ is a predetermined matrix and $y \in \mathbb{R}^d$. To successfully solve this task, an attention-based model must learn to attend to the last token only and learn the ground-truth target weights $W^*$. We deliberately incorporate a sparse attention target mechanism to study the relationship between sparse attention and emergence. This task shares similarities with the

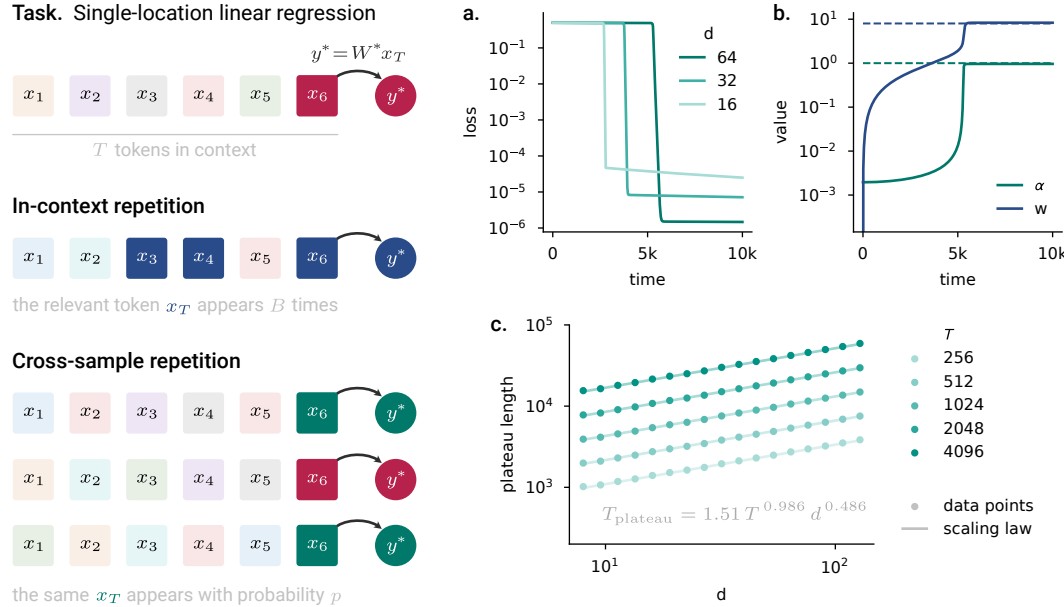

Figure 1: **A simple task to study the emergence of sparse attention.** (left) We introduce a variant of linear regression task that is analytically tractable and in which Transformers-like models need to learn sparse attention. The model must identify which token (here the last one, $x_T$) is relevant for the target output $y^*$. We incorporate two realistic forms of repetition in the data: in-context repetition, where the relevant token appears multiple times within the context, and cross-sample repetition, where an input sequence contains a special token $\tilde{x}$ (here colored in green) at the relevant position with probability $p$. See Section 2.1 for details. (right) a. As desired, the reduced learning dynamics of our simplified Transformer (Eq. 2) exhibit a multi-phase behavior including an initial plateau, on the task without repetition ($T = 512$). b. Mechanistically, the weights $w$ begin learning before attention to the relevant token $\alpha$ ($T = 512, d = 64$). Dashed lines represent optimal values. c. The duration of the initial plateau increases as a function of sequence length $T$ and input/output dimension $d$, closely following a power law scaling relationship ($R^2 = 0.999$) that can be accurately predicted by linearizing the dynamics around initialization (Equation 6). See Section 2.2 for details.

single-location regression task of Marion et al. [27], although in our case, the relevant token location is always the same. The left panel of Figure 1 summarizes this task.

We model two forms of repetition that are ubiquitous in natural language:

- **In-context repetition.** This occurs when specific token groups (e.g., a person's name) repeatedly appear within a single context window. In our framework, we model this property by repeating the relevant token $x_T$ multiple times in the sequence $(x_t)_{t=1}^T$, always at the same positions for our simplified model to be able to use it. Following Chan et al. [13], who termed this property "burstiness", we denote $B$ as the number of times the task-relevant token appears in the context.

- **Cross-sample repetition.** This form of repetition comes from having certain information (such as biographical details of specific individuals) overrepresented in the overall training data. We implement this by first sampling the input sequence normally, but then occasionally (with probability $p$) replacing the relevant token $x_T$ with a special predefined token $\tilde{x}$.

For the purpose of our theoretical analysis, we introduce a simplified attention layer, defined by

$$y = W \sum_{t=1}^{T} \text{softmax}(a)_t \, x_t \tag{2}$$

with $a \in \mathbb{R}^T$ the attention scores vector and $W \in \mathbb{R}^{d \times d}$ the weight matrix. Unlike in standard Transformer attention [28], our model does not use any semantic information to determine where to attend. This simplification facilitates theoretical analysis and implicitly assumes that the attention

layer has already learned to filter irrelevant contextual information. The model's parameters $(a, W)$ are learned by minimizing the expected mean square error between predictions $y$ and targets $y^*$.

## 2.2 The learning dynamics of the simplified Transformer exhibit emerging abilities

The performance of our simplified model on this task exhibits a characteristic learning pattern: an initial plateau where the loss minimally decreases, followed by a sharp phase transition towards significantly lower loss values. The analysis we detail below reveals that this emergent behavior arises from the interaction between feedforward weights and attention during learning, and that the duration of the initial plateau increases with the sequence length $T$ and data dimensionality $d$.

**Reduced learning dynamics.** We analyze the gradient flow dynamics of the simplified model. The assumptions and mathematical details of this analysis can be found in Appendix A. Under these mild assumptions, the learning dynamics of the entire model reduces to two key scalar variables: $\Delta a := a_T - a_t$ for any $t < T$ (all attention scores to non-relevant tokens stay the same under our assumptions) and $w$ the scalar projection[1] of $W$ on $W^*$. These variables are initially both equal to $0$ and they evolve according to the following system of ordinary differential equations:

$$\dot{w} = \frac{\alpha(\sqrt{d} - \alpha w)}{d} - \frac{(1 - \alpha)^2 w}{d(T - 1)} \tag{3}$$

$$\dot{\Delta a} = \alpha(1 - \alpha)\left(\frac{w(\sqrt{d} - \alpha w)}{d} + \frac{(1 - \alpha)w^2}{d(T - 1)}\right), \tag{4}$$

with $\alpha := (1 + (T - 1)\exp(-\Delta a))^{-1}$ the attention given to the final token. In these two equations, the first term is the signal coming from the relevant token at position $T$, and the second term is the noise coming from the $T - 1$ non-relevant tokens. This decomposition follows from the derivation in Appendix A.3, which groups token contributions by their relevance to the target prediction.

These equations enable us to elucidate the roots of emergence in this task. Initial learning is slow, as $\Delta a$ does not receive any teaching signal as $w = 0$ and $w$ slowly increases as attention is initially uniform ($\alpha = 1/T$). As a consequence, the feedforward weight $W$ must first align with the target weights $W^*$ before attention can learn. A positive feedback loop is then progressively established: increased attention improves the learning signal for $w$, and a better-learned $w$ further reinforces the correct attention pattern. This dynamic, similar to the one found in deep linear networks [29], leads to a sharp decrease in loss and the sudden emergence we observe in Figure 1.a.

**Predicting when emergence arise.** To estimate how long it takes to escape the initial loss plateau, we linearize the dynamics around the initial conditions and obtain

$$\begin{pmatrix} \dot{w} \\ \dot{\Delta a} \end{pmatrix} = \begin{pmatrix} \frac{1}{\sqrt{d}T} \\ 0 \end{pmatrix} + \begin{pmatrix} 0 & \frac{1}{\sqrt{d}T} \\ \frac{1}{\sqrt{d}T} & 0 \end{pmatrix} \begin{pmatrix} w \\ \Delta a \end{pmatrix}. \tag{5}$$

This linearization provides two key insights: First, it confirms that feedforward weight learning precedes and drives attention learning, as evidenced by the initial gradient and the top-right entry of the matrix in the equation above, and corroborated by the simulations of Figure 1.b. Second, it enables us to estimate the escape time from initial conditions, defined as the time required to reach $(1 - \varepsilon)$ of the initial loss value. It is equal to

$$T_\varepsilon = \frac{\sqrt{d}T}{2} \ln\left(\varepsilon\sqrt{d}T\right). \tag{6}$$

This formula succinctly demonstrates that both longer sequences and higher-dimensional inputs increase the time spent on the plateau and delay emergence. This theoretically derived scaling closely matches the one obtained in simulations, as depicted in Figure 1.c ($\varepsilon = 0.8$ in these simulations).

## 2.3 Repetition speeds up emergence

Now that we have thoroughly examined the vanilla case, we focus on understanding the effects of repetition. Our analysis reveals that in-context repetition makes the attention pattern to be learned less

---

[1] $w$ is formally defined as $w := \langle W^*, W \rangle_F / \|W^*\|_F \in \mathbb{R}$ with $\langle \cdot, \cdot \rangle_F$ the Froebenius inner product.

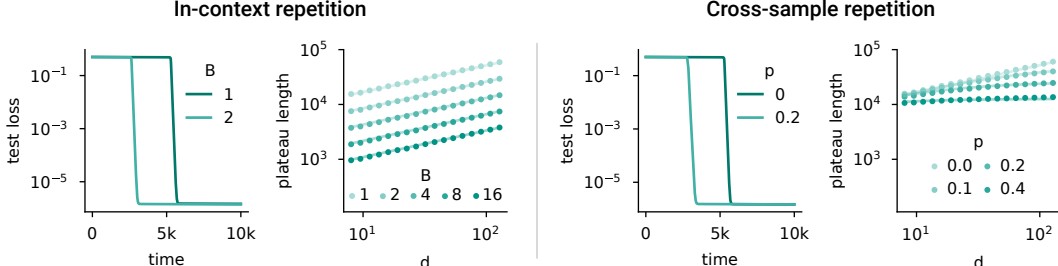

Figure 2: **Repetition speeds up emergence in the linear regression task in a theoretically predictable way.** (left) Increasing in-context repetition through $B$ reduces the initial plateau, and the length of the plateau is well captured by the power law $T_{\text{plateau}} = 1.51\, T^{0.99} B^{-0.99} d^{0.49}$ ($R^2 = 0.999$). (right) Cross-sample repetition, modulated by the repetition probability $p$, exhibits similar effects, even when evaluating the model on a test loss without repetition (i.e., $p = 0$). The length of the plateau follows $T_{\text{plateau}} = 2.15(\sqrt{d}T/\sqrt{p^2 d + (1-p)^2})^{1.02}$ ($R^2 = 0.992$). The plateau length in both cases closely follows theoretical predictions. See Section 2.3 for details.

sparse, thereby simplifying the task, while cross-sample repetition accelerates feedforward weight learning in specific directions, which subsequently increases overall learning speed by increasing the attention of the model to relevant tokens earlier. We present the key findings below.

**In-context repetition.** The role of in-context repetition is rather simple. When relevant information repeats multiple times within the context, it becomes more correlated with the token to predict. This can be understood as increasing the signal to noise ratio or as effectively reducing the sequence length from $T$ to $T/B$. Since learning time scales with sequence length, this makes the task fundamentally easier. Theoretically, we can show, cf. Appendix A.3, that this intuition precisely holds and that the escape time from the initial loss plateau becomes proportional to $T\sqrt{d}/B$. Replacing $T$ by $T/B$ in the scaling law we obtained in Figure 1.c yields an almost perfect empirical fit, cf. Figure 2.b. This result highlights that $B$ reduces the sparsity of the target attention pattern and thus accelerates emergence.

This analysis ignores, by design, any ability of the attention mechanism to flexibly use semantic or positional information, as we directly parameterized the attention scores. We argue that our findings will extend to the more general case. Indeed, both the attention scores, which would now be the output of some function, and the feedforward weights receive larger gradients with in-context repetition, and thus learning will overall be faster. We will confirm that the same conclusion holds empirically for more realistic architectures and optimizers in the next sections.

**Cross-sample repetition.** The role of cross-sample repetition is more intricate. Repeating one token more frequently, that is increasing $p$, causes the input covariance matrix of the relevant token, $\mathbb{E}[x_T x_T^\top]$, to become anisotropic. The different components of the weight matrix $W$ are then learned at different speeds. While this difference in learning speed traditionally leads to slower convergence rates in vanilla linear regression [30], it turns out to be beneficial in our attention-based version of the task. Indeed, following similar principles to the ones detailed in the previous section, learning weights in the repeated direction will lead to the attention to the relevant tokens increasing faster, and this then speeds up the learning of the non-repeated dimensions. As a consequence, the model escapes the initial learning plateau faster. Importantly, this also holds when measuring the model's performance on non-repeated data, as seen in Figure 2. The effect of cross-sample repetition can also be understood as increasing the signal-to-noise ratio. Repeating the same token increases the amount of signal coming from the relevant token while keeping the amount of noise received from other tokens fixed, at the price of losing some information about the relevant signal.

Theoretical analysis requires the introduction of an additional variable, so that the learning speed in both the repeated dimension and the other dimensions can be tracked independently. This model, which we introduce in Appendix A.4, enables us to justify the mechanistic insights mentioned above, as well as to derive that the plateau length scales as $\sqrt{d}T/\sqrt{p^2 d + (1-p)^2}$, which accurately describes empirical behavior (cf. Figure 2). The repetition probability $p$ thus implicitly interpolates

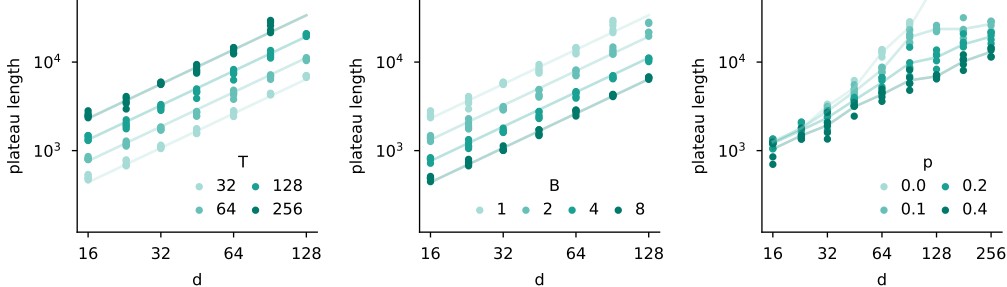

Figure 3: **Theoretical insights on the linear regression task transfer to more realistic versions of the task, the model, and the optimizer.** Transformer-based architectures exhibit similar phase transitions as our toy model and its corresponding plateau length follows similar trends to the ones derived in theory. (left and middle) Evolution of the plateau length as a function of $d$, when varying $T$ and $B$ (by default $T = 256$ and $B = 1$). The lines corresponds to the power law $T_{\text{plateau}} = 0.76\,d^{1.29}\,T^{0.80}\,B^{-0.80}$ ($R^2 = 0.995$). (right) Same plot, this time varying the cross-sample repetition probability $p$. The lines correspond to the evolution of the average plateau length as $d$ increases, for the different $p$ values. See Section 2.4 for details.

between the $d$-dimensional case and the 1-dimensional case, highlighting that the learning of the feedforward mapping becomes less of a bottleneck.

However, cross-sample repetition is not a free lunch: it only provides a temporary advantage. First, these dynamics only have a smaller test loss in the medium term. In the long run, learning is slower for similar reasons as for the standard linear regression. Second, there is an additional overfitting problem. While it does not appear in this simplified setting as anisotropic input covariance does not bias the final solution, it starts appearing for more realistic architectures. This phenomenon has been observed in practice: examples include [31, 16, 18] for synthetic tasks that have properties similar to the one we focus on here, and [32, 33] for the pretraining of large language models.

## 2.4 The theory qualitatively captures learning dynamics of full-fledged Transformers

We conclude our linear regression analysis by examining how our theoretical predictions extend to more realistic task versions, optimizers, and models – particularly those with standard attention that varies with inputs. Our findings, detailed in the remainder of this section, indicate that learning dynamics behave qualitatively similarly, showing sharp phase transitions, with emergence time maintaining similar dependencies on data properties. However, the precise dependencies differ, which we investigate in more depth.

For this investigation, we train a standard 2-layer 4-heads Transformer with the Adam optimizer [34], using a constant learning rate of $10^{-4}$. Our only deviation from standard architectures is removing layer normalization, which makes solving the task more challenging. To enhance task realism, we randomly sample the positions of relevant tokens and incorporate an additional feature into the input to indicate whether a token is task-relevant. Further experimental details are provided in the appendix.

The learning dynamics of Transformers align qualitatively with our theoretical predictions. First, the loss evolution exhibits a sharp phase transition (see Figure 11 in the appendix for an example). Second, emergence time depends similarly on the data properties identified in our theory, with repetition accelerating emergence. However, this result comes with several nuances.

For scenarios with no repetition or with in-context repetition, power laws still accurately capture the dependency of emergence time on sequence length $T$, dimension $d$ and burstiness $B$, though with significantly different exponents. Ablations (see Appendix B.3) reveal that optimizer, architecture, and task specifics all influence these exponents. Notably, replacing Adam with SGD substantially slows emergence, both in absolute terms and by increasing dependency on task difficulty. This finding illustrates a broader observation: Adam is crucial for efficient Transformer learning [e.g., 35].

Regarding cross-sample repetition, power laws no longer accurately describe the empirical relationship, partly because measuring plateau length becomes more difficult (see the learning dynamics

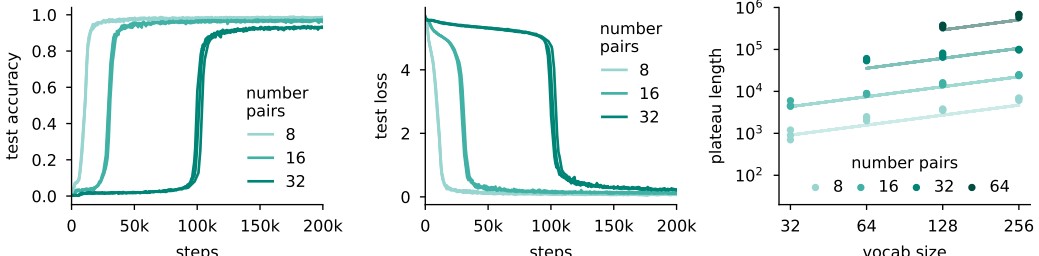

Figure 4: **In-context learning emerges in the associative recall task, and emergence time grows with the number of pairs in the context and the vocabulary size.** (left and middle) The ability to solve the task emerges through training, with emergence time increasing as the task gets harder (by increasing the number of pairs here, the vocabulary size being fixed to $256$). This is qualitatively consistent with the theory developed for sparse attention. (right) Systematically investigating the relationship between emergence time and data properties reveals that it follows the power law $T_{\text{plateau}} = 0.55 \, N_{\text{tokens}}^{0.79} \, N_{\text{pairs}}^{2.25}$ ($R^2 = 0.982$). Results are only shown when the number of pairs is larger than the vocabulary size, to ensure that the query does not appear multiple times in the context (we do not consider repetition here). More details, in particular a hypothesis for why the $N_{\text{pairs}}$ exponent is so high, can be found in Section 3.2.

under cross-sample repetition in Figure 12 in the appendix for an example). Nevertheless, the trends identified in our theory still hold: this form of repetition accelerates emergence, with the effect becoming more pronounced as $d$ increases.

## 3 Emergence of in-context learning, through the lens of sparse attention

We conclude by investigating to what extent the insights developed on our simple linear regression task extend to more realistic learning problems, in particular one in which in-context learning emerges. To this end, we examine an in-context associative recall task, which can be solved by learning an induction head – a well-studied circuit strongly implicated in in-context learning [12, 15], which necessitates at least two attention layers with sparse attention patterns. Our theory qualitatively captures both the learning dynamics and the impact of the training distribution on learning speed.

### 3.1 The in-context associative recall task

The in-context associative recall task serves as a standard benchmark for testing language models' ability to access information in their context and to perform a simple form of one-shot learning. This task requires abilities that correlate with models' capacity for in-context learning [12], and variations of it have been extensively studied to better understand how in-context learning emerges [13, 15, 36–40]. It has also been used as a testbed for recently proposed linear recurrent architectures [e.g. 41–43], as it acts as an important differentiator between attention-based and recurrent architectures [44].

We implement this task as follows: each sequence consists of $N_{\text{pairs}}$ key-value token pairs (5 such pairs in the example below) followed by a query token (Z below), as shown:

$$\text{Y I \quad A X \quad U R \quad Z Y \quad C A \quad Z ?}$$

The query corresponds to one of the keys in the context, and the model must output the corresponding value – Y in the example above (since the query Z matches the key in the pair Z Y). In practice, we work with a total of $N_{\text{tokens}}$ unique tokens provided to the network via one-hot encodings. We train the model using a cross-entropy loss comparing the model output to the target value.

The number of pairs $N_{\text{pairs}}$ controls the sequence length $T$ as $T = 2N_{\text{pairs}} + 1$ (accounting for the query token), and the vocabulary size $N_{\text{tokens}}$ plays a role comparable to the input dimension $d$ in the linear regression task. To model in-context repetition, we ensure that the query token appears on average $B$ times in the context (as a key). To model cross-sample repetition, we select a subset of 2 tokens that will appear more often[2] and vary $p$, the probability to sample the query from this subset.

---

[2]2 is an arbitrary choice that we have not ablated.

The precise description of the task is provided in the appendix. The results we report in the main text are testing the learned models on data without any repetition, thereby testing the generalization abilities of models learned on repeated data.

Transformers typically solve this type of task by implementing an induction head – a circuit that combines an attention layer responsible for concatenating the representation of the current token with the previous token, and a selection attention layer responsible for retrieving the relevant information from the context. Both attention layers focus on a sparse subset of tokens (usually just one), making this task particularly well-suited to test our theory. Unlike the linear regression task where sparse attention was hard-coded in the target input-output mapping, there is no inherent constraint forcing models to develop sparse attention here. We note that our toy model (Equation 2) cannot directly implement an induction head, as it lacks the relative positional information for copying and semantic similarity mechanisms for token matching. This task thus serves as the perfect testbed to show that our theoretical insights extend to more realistic, yet still finely controlled, learning scenarios in which their simplifying assumptions do not hold.

### 3.2 The emergence of in-context learning depends on data as in the theory

As with the linear regression task, we investigate the learning dynamics of Transformers on this task and compare the qualitative findings with those of our developed theory. The experimental setup being closely related to the one in Section 2.4, we defer its thorough description to the appendix and note that we use layer normalization and MLPs for this set of experiments.

Overall, we find that our theory accurately describes both the learning dynamics (in-context learning emerges in this task) and the dependency of emergence timing on data distribution properties.

**Longer sequences and larger vocabulary size delay emergence.** We begin by verifying that the intuition described above applies and that the learning curve exhibits the sharp phase transitions characteristic of emergence. This is indeed the case for sufficiently long sequences and/or large vocabularies, as illustrated in Figure 4. Importantly, there is only one phase transition despite two attention layers being needed to learn sparse patterns; Figure 17 shows that they are learned simultaneously. This observation is consistent with the results of Reddy [36] and Singh et al. [15].

In Figure 4 (right), we report how the emergence time, arbitrarily defined as the time needed to reach $5\%$ accuracy, evolves as a function of sequence length and data dimension. It accurately follows a power law, as in all our other experiments, albeit with exponents that are relatively low for $N_{\text{tokens}}$ (0.79) and extremely high for $N_{\text{pairs}}$ (2.25). We hypothesize that the lower exponent for $N_{\text{tokens}}$ stems from the fact that most of the feedforward processing can be handled by residual connections, and given that the dimension of the data primarily influences the speed of feedforward learning, data dimension does not have such a large effect. For the higher exponent for $N_{\text{pairs}}$, our hypothesis is that this occurs as two sparse attention patterns must be learned jointly. In the toy model we analyze theoretically, a similar coupling exists between the attention and the feedforward weights, resulting in a multiplicative interaction. We posit that a similar mechanism may be at work here, with each attention layer contributing additively to the $N_{\text{pairs}}$ exponent. We leave a more thorough investigation of how different circuits interact to influence emergence time to future work.

**Repetition speeds up emergence but comes with overfitting risks.** We next investigate the benefits of in-context and cross-sample repetition. From the high sensitivity of the emergence time on sequence length revealed by previous analysis, we expect in-context repetition to have a stronger effect than the cross-sample one. This is what we observe. The results reported in Figure 5, which evaluate the model performance on test data that has no repetition, demonstrate that repetition can significantly speed up emergence, by a factor of 4 for in-context repetition and a factor of 2 for cross-sample repetition. The attention sparsity perspective thus explains why in-context repetition has been found to favor in-context learning vs. in-weight learning [13, 36, 45]: the induction head needed to solve this kind of task is formed earlier with such a form of repetition.

However, this benefit comes with an overfitting risk: while repetition consistently accelerates learning (repetition always speeds up learning on the train loss, cf. Figure 16 in the appendix), too much repetition leads to learning strategies that do not generalize well. For example, selecting the most frequent value is a valid strategy for this task whenever the query appears two or more times in the context. Interestingly, we also observe some grokking-like patterns [46] as the test accuracy eventually starts to increase late in training for large amounts of repetition. We argue that this occurs

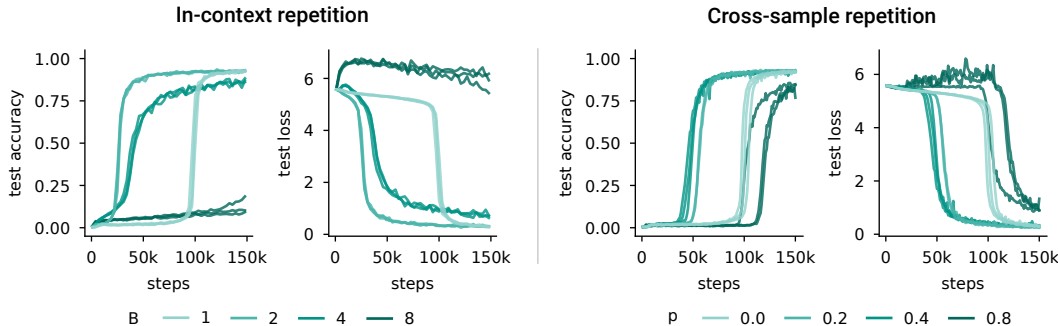

Figure 5: **Repetition speeds up emergence in the in-context associative recall task but comes with overfitting risks.** (left) We vary the amount of in-context repetition $B$, that is the number of times the query appears as a key in the context on average, and find significant benefits of small amount of repetition. Larger amounts of repetition lead to overfitting, but learning for long enough eventually leads to grokking. (right) Cross-sample repetition, more precisely the probability $p$ that the query is one of the 2 repeated tokens, has similar effects. Results are obtained for $N_{\text{pairs}} = 32$ and $N_{\text{tokens}} = 256$.

because the no-repetition case is still represented in the training data, albeit with very little weight. This trade-off between learning speed and generalization is consistent with what Park et al. [31] reported on another in-context learning task. The overfitting we observe here may appear inconsistent with our theory but is not: our theory addresses learning speed (performance on the training loss) rather than generalization ability.

## 4   Discussion

**Connection with other theoretical work.**   Through the lens of sparse attention, we provide a unifying perspective on a set of empirical results highlighted in the motivation (Section 1). While our focus on sparse attention and the effect of repetition is unique (to the best of our knowledge), there are multiple closely related works. First, the interaction between the feedforward weights and attention we study is closely related to the one between two consecutive feedforward linear layers [29, 47, 48]. Linear networks also display incremental learning with abrupt transitions characterizing each phase change [49–52]. This line of research on linear networks has since then been extended to linear [45, 53] and softmax [54] Transformers, in particular to study the emergence of in-context learning. On the more conceptual side, there exist other theories of how emergence could arise from longer training, such as grokking [46, 14] or singular learning theory [55, 56]. While rarer, other theories focus on emergence from increased model size [57]. Compared to these, our theory focuses on learning dynamics (emergence over the course of training) and is directly tied to the internal mechanisms of the Transformers.

**How much emergence comes from sparse attention?**   We find that the learning of sparse attention is prone to producing emergence over the course of training. Given that sparse or concentrated attention is a common emergent feature of Transformers (e.g., induction heads [12], task/function vectors [58, 59], or high-norm tokens in vision transformers [60]), one may ask how much currently known emergent behaviors can be attributed to the learning of sparse attention patterns, and whether there are many more emergent behaviors than what is currently reported. However, these points must be weighed. As noted above, even simpler MLPs can give rise to abrupt emergence over training [e.g. 29]; thus, learning in other circuits within Transformers might contribute to sudden transitions in their performance. Furthermore, emergence results at large scale mostly show a sharp phase transition of the model as the number of FLOPs [e.g., 8], which roughly corresponds to the model size times the number of training steps, reaches a certain threshold. It is therefore unclear whether these examples of emergence are rooted in longer training, models with higher capacity, or a complex interplay between the two. More empirical and theoretical work is needed to better understand the causes of emergence in large models and the possible complex relationship between training-based and size-based emergence.

**When does repetition help?** Overall, we expect repetition to be beneficial when training on tasks prone to performance plateaus, as there is a clear benefit in trading off learning speed for generalization in these cases. Cross-sample repetition should be particularly helpful in tasks where learning feedforward transformations is challenging (high $d$ in our analysis). This includes learning many factual associations as in Zucchet et al. [18] and the reasoning-heavy arithmetic tasks of Charton and Kempe [16], which require learning complex non-linear transformations. In-context repetition should be most beneficial when learning from very long sequences, as it effectively reduces the sequence length the model must process. Repetition, both in-context and cross-sample, is a natural feature of language that may accelerate some parts of language learning. The principles we touch upon could already be more directly at play in the current training pipeline of large language models, for instance when code, which typically has lower syntactic diversity than other types of text [61], is included at specific moments during training.

**Data diversity as a path towards active learning?** A key result of our work is that low data diversity can actually improve performance. This contrasts with classic machine learning principles, which state that low diversity hurts generalization. These two apparently conflicting statements are actually compatible. As our results highlight, low diversity initially accelerates the learning of sparse attention, but high data diversity is better as training time goes to infinity. We hypothesize that data diversity might be a powerful lever towards enabling active learning [62]. Specifically, our findings suggest a simple active learning algorithm: whenever an agent detects it is stuck on a task, it can decrease data diversity to accelerate learning, then increase diversity after the critical transition occurs to improve generalization. This dynamic adjustment of diversity provides a practical mechanism for agents to actively control their own learning trajectories, with the ultimate goal of letting the learner decide what it wants to learn on and reaching human-level sample efficiency that current deep learning systems crucially lack [63–65]. Humans also encounter varying levels of data diversity over development. For example, infants receive progressively increasing data diversity during their early years [66]. Our theory suggests that it may be an important factor in accelerating our development. While promising, this thread requires more extensive analysis, both at larger scale and on more realistic data distributions.

## Acknowledgments

The authors thank Jay McClelland for discussions that inspired this work. F.D. thanks Aditya Varre for the useful discussions.

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

# Appendix

## Table of contents

# A  Details of the theoretical results

## A.1  Overview of the assumptions

Throughout our theoretical analysis, we introduce a few assumptions to simplify it. We summarize them here:

Asm. 1 **Gradient flow dynamics.** To study learning dynamics, we focus on the gradient flow of the expected loss. Compared to the standard deep learning regime, this removes stochastic noise induced by sampling, uses gradient descent as optimizer instead of optimizer with adaptive learning rates such as Adam, and requires an infinitely small learning rate. Due to the last point, phenomena such as the edge of stability [67] are out of the picture. Yet, this is a very standard assumption (e.g. [29]) and will enable us to use tools from dynamical systems to understand how the behavior of the model changes over the course of learning.

Asm. 2 **Uniform attention at initialization.** At initialization, the attention patterns of Transformers generally display remarkable uniformity at initialization. We take advantage of this observation in our model by initializing $a$ as a vector of zeros. This enables us to reduce the dynamics of the attention part of the model to a single scalar.

Asm. 3 $W^*$ **has norm 1 columns.** We assume that the columns of $W^*$ all have norm 1. While this is not strictly necessary for performing our analysis, it simplifies the formulas that we obtain and helps to keep the notation concise. Additionally, this is the expected behavior when drawing the entire of $W^*$ i.i.d. from a zero-mean normal distribution with variance $\frac{1}{d}$ when $d$ goes to infinity.

Asm. 4 **Initialize the weights $W$ at 0.** When drawing the entries of $W$ i.i.d. and independently of those of $W^*$, $W$ is almost surely orthogonal to $W^*$ as $d \to \infty$. The components of $W$ not aligned with $W^*$ exhibit a fast decay towards 0, so this assumption corresponds to having already completed this process. With such an assumption, we can reduce the dynamics of the weights to a single scalar, $w$, which will be the projection of $W$ on a normalized $W^*$.

Asm. 5 **Large sequences and large dimension.** Finally, we assume that we are in the large $T$ limit and $B$ is negligible in front of $T$. This makes sense in our context as we are interested in modeling the learning of sparse attention patterns. We additionally assume that $d$ is large, which is a reasonable assumption given that we are interested in modeling the high-dimensional data neural networks have to process. These assumptions will enable us to ignore some negligible terms and keep our calculations more concise.

## A.2  Derivation of the expected loss and its gradients

Recall that the loss is given by

$$L = \frac{1}{2}\mathbb{E}_x\left[\|y - y^*\|^2\right].$$

with

$$y = W\sum_{t=1}^{T} \text{softmax}(a)_t\, x_t$$

and

$$y^* = W^* x_T.$$

For notational clarity, we define attention patterns as $\alpha_t := \text{softmax}(a)_t$. Without loss of generality, we assume that the tokens repeated in tokens appear at positions $\{T - B + 1, \cdots, T\}$ so that the last $B$ tokens are always the same, and that the repeated input $\tilde{x}$ is the first vector of the canonical basis[3] of $\mathbb{R}^d$, that is $\tilde{x} = [1, 0, \cdots 0]^\top$. For the sake of conciseness, we denote by

$$\Sigma_t := \mathbb{E}_x\left[x_t x_t^\top\right]$$

the input covariance of each token. Note that for $t \leq T - B$, it is equal to $\Sigma_t = \frac{1}{d}\text{Id}$.

---

[3] This amounts to a right multiplication of the weights $W$ by an orthogonal matrix.

Using the insights mentioned above and the properties of the data distribution, the loss reduces to:

$$L = \frac{1}{2}\mathbb{E}_x\left[\|y - y^*\|^2\right]$$

$$= \frac{1}{2}\mathbb{E}_x\left[\left\|\sum_t \alpha_t W x_t - W^* x_T\right\|^2\right]$$

$$= \frac{1}{2}\mathbb{E}_x\left[\sum_{t \leq T-B} \alpha_t^2 \|W x_t\|^2 + \left\|\sum_{t>T-B} \alpha_t W x_t - W^* x_T\right\|^2\right]$$

$$= \frac{1}{2}\mathbb{E}_x\left[\sum_{t \leq T-B} \alpha_t^2 \|W x_t\|^2 + \left\|\sum_{t>T-B} \alpha_t W x_T - W^* x_T\right\|^2\right]$$

$$= \frac{1}{2}\left[\sum_{t \leq T-B} \alpha_t^2 \mathrm{tr}\left(W\Sigma_t W^\top\right) + \mathrm{tr}\left(\left(\sum_{t>T-B} \alpha_t W - W^*\right)\Sigma_T\left(\sum_{t>T-B} \alpha_t W - W^*\right)^\top\right)\right]$$

$$= \frac{1}{2}\left[\sum_{t \leq T-B} \frac{\alpha_t^2}{d} \|W\|_F^2 + \mathrm{tr}\left(\left(\sum_{t>T-B} \alpha_t W - W^*\right)\Sigma_T\left(\sum_{t>T-B} \alpha_t W - W^*\right)^\top\right)\right].$$

In the third line, we used the fact that the first $T - B$ tokens are independent of each other and independent of the last $B$ ones, in the fourth line that the last $B$ tokens are all equal to $x_T$ and in the fifth line the equality $\mathbb{E}_x[\|Ax\|] = \mathrm{tr}(A\mathbb{E}[xx^\top]A^\top)$.

Leveraging this formula, we get the following gradients for the loss:

$$\nabla_W L = \sum_{t>T-B} \alpha_t \left(\sum_{t>T-B} \alpha_t W - W^*\right)\Sigma_T + \sum_{t \leq T-B} \frac{\alpha_t^2}{d} W \qquad (7)$$

$$\nabla_{\alpha_t} L = \begin{cases} \mathrm{tr}\left(W\Sigma_T\left(\sum_{t>T-B} \alpha_t W - W^*\right)^\top\right) & \text{if } t > T - B \\ \frac{\alpha_t}{d}\|W\|^2 & \text{otherwise.} \end{cases} \qquad (8)$$

Note that we have not calculated the gradients with respect to $a$ here, but only with respect to $\alpha$.

### A.3 Analysis of the dynamics without cross-sample repetition

In this section, we study the dynamics without cross-sample repetition, that is, with $p_{\text{repeat}} = 0$.

#### A.3.1 Reducing the learning dynamics to two dimensions

In general, the parameters evolve in a high-dimensional space. However, with a few reasonable assumptions, it is possible to show that they evolve in a 2-dimensional subspace:

– **Attention is initialized uniformly.** The first assumption we make is that attention is perfectly uniform (Assumption 2 from Section A.1), that is $\alpha_t = \frac{1}{T}$ or $a_t = 0$. Given that the gradients of these parameters are the same for $t \leq T - B$ and $t > T - B$, cf. the calculation in the previous section, all attention values will have two possible values. If we additionally leverage the fact that $\sum_t \alpha_t = 1$, everything is captured by a single scalar $\Delta a$, that is defined as $a_T - a_1$. Indeed

$$\alpha_t = \frac{\exp(a_t)}{\sum_{t'} \exp(a_{t'})}$$

$$= \frac{1}{(T - B)\exp(a_1 - a_t) + B\exp(a_T - a_t)}$$

so that, for $t > T - B$,

$$\alpha_t = \frac{1}{(T - B)\exp(-\Delta a) + B} \qquad (9)$$

and for $t \le T - B$,

$$\alpha_t = \frac{(1 - B\alpha_T)}{(T - B)}.$$ (10)

– **Weights are initialized at** $0$. We assume that $W = 0$ at initialization, following Assumption 4 of Section A.1. As $\Sigma_T = \frac{1}{d}\text{Id}$, we have

$$\nabla_W L = \sum_{t > T-B} \frac{\alpha_t}{d} \left( \sum_{t > T-B} \alpha_t W - W^* \right) + \sum_{t \le T-B} \frac{\alpha_t^2}{d} W.$$

This implies that whenever $W$ is aligned with $W^*$, which is the case when $W = 0$, it remains aligned for the rest of learning. Yet, we are not entirely done as the precise parametrization is important to ensure that the dynamics match. Such a property is achieved when $W = wW^*/\|W^*\|_F$ as, under the gradient flow dynamics on $W$, we have

$$w = \frac{\langle W^*, W \rangle_F}{\|W^*\|_F}$$

$$\dot{w} = \frac{\langle W^*, \dot{W} \rangle}{\|W^*\|_F} = -\frac{\langle W^*, \nabla_W L \rangle_F}{\|W^*\|_F}$$

and under the gradient flow dynamics directly on $w$, we get

$$\dot{w} = -\nabla_w L = -\left\langle \frac{dW}{dw}, \nabla_W L \right\rangle_F = -\frac{\langle W^*, \nabla_W L \rangle_F}{\|W^*\|_F}.$$

In the calculations above, we used $\langle \cdot, \cdot \rangle_F$ to denote the element-wise dot product (i.e., $\langle A, B \rangle_F = \sum_{ij} A_{ij} B_{ij}$). The corresponding norm is the Froebenius norm $\|\cdot\|_F$.

– **Each column of the target weights** $W^*$ **has norm 1.** It follows that the Froebenius norm of $W^*$ satisfies $\|W^*\|_F^2 = d$. This is Assumption 3 from Section A.1.

Under these assumptions, studying the gradient flow dynamics on the original loss therefore reduce to study the gradient flow dynamics on the simplified loss

$$L = \frac{1}{2} \left[ \sum_{t \le T-B} \frac{(1 - B\alpha)^2}{d(T-B)^2} \frac{w^2}{\|W^*\|_F^2} \|W^*\|_F^2 + \frac{1}{d} \left( \frac{B\alpha w}{\|W^*\|} - 1 \right)^2 \text{tr}\left(W^* W^{*\top}\right) \right]$$ (11)

$$= \frac{1}{2} \left( \frac{(1 - B\alpha)^2 w^2}{d(T-B)} + \frac{\left(B\alpha w - \sqrt{d}\right)^2}{d} \right)$$ (12)

with the attention given to token $T$ being equal to

$$\alpha = \frac{1}{(T-B)\exp(-\Delta a) + B}.$$

Note that we have dropped the $T$ subscript for notational conciseness. We plot this reduced loss landscape in Figure 6, as well as how the parameters evolve under the gradient flow dynamics.

As $d_{\Delta a}\alpha = \alpha(1 - B\alpha)$, the gradient of this loss with respect to $w$ and $\Delta a$ are

$$\nabla_w L = \frac{(1 - B\alpha)^2 w}{d(T-B)} + \frac{B\alpha(B\alpha w - \sqrt{d})}{d}$$ (13)

$$\nabla_{\Delta a} L = \alpha(1 - B\alpha) \left( -\frac{B(1 - B\alpha)w^2}{d(T-B)} + \frac{Bw(B\alpha w - \sqrt{d})}{d} \right).$$ (14)

The entries of the Hessian are

$$\frac{d^2 L}{dw^2} = \frac{(1 - B\alpha)^2}{d(T-B)} + \frac{B^2\alpha^2}{d}$$

$$\frac{d^2 L}{dw\,d\Delta a} = \alpha(1 - B\alpha) \left( -\frac{2B(1 - B\alpha)w}{d(T-B)} + \frac{B(2B\alpha w - \sqrt{d})}{d} \right)$$

$$\frac{d^2 L}{d\Delta a^2} = \frac{1 - 2B\alpha}{\alpha(1 - B\alpha)} \nabla_{\Delta a} L + \alpha(1 - B\alpha) \left( \frac{B^2 w^2}{d(T-B)} + \frac{B^2 w^2}{d} \right).$$

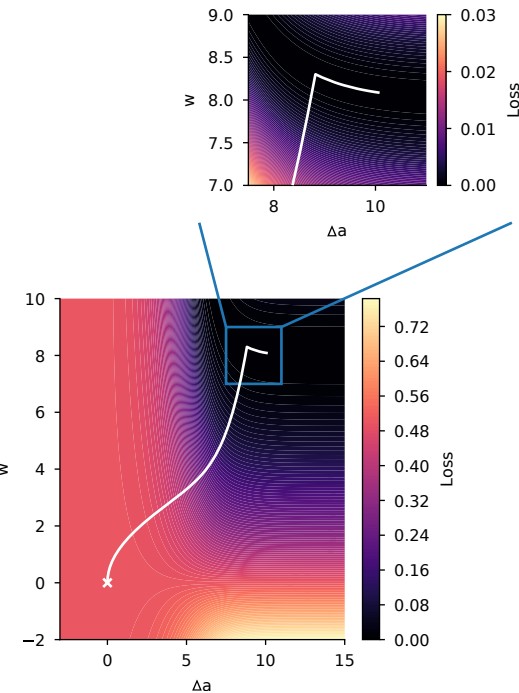

Figure 6: Loss landscape for the reduced model ($T = 256$, $d = 64$, without any repetition). The white line corresponds to the gradient flow on this loss, initialized in $(0, 0)$ (white cross).

### A.3.2 Approximate dynamics around initialization

Despite having reduced the dynamics to two dimensions, it is still too complex to be analyzed mathematically. In order to gain insight into the early behavior of the dynamics, we linearize the dynamics around initialization and compute the time it will require to leave the initial loss plateau.

**Linearized dynamics at initialization.** Linearizing the dynamics at initialization corresponds to considering the gradient flow on the quadratic approximation to the loss:

$$L(w, \Delta a) = L(0,0) + \nabla L(0,0)^\top \begin{pmatrix} w \\ \Delta a \end{pmatrix} + \frac{1}{2} \begin{pmatrix} w \\ \Delta a \end{pmatrix}^\top H(0,0) \begin{pmatrix} w \\ \Delta a \end{pmatrix} + o(\|w\|^2 + \|\Delta a\|^2),$$

with $\nabla L(0,0)$ the gradient of the loss and $H(0,0)$ the Hessian of the loss at initialization. Plugging in the values the different variable take at initialization gives

$$\nabla_w L = -\frac{B}{\sqrt{dT}}$$

$$\nabla_{\Delta a} L = 0$$

and

$$\frac{\mathrm{d}^2 L}{\mathrm{d}w^2} = \left( \frac{T - B}{dT^2} + \frac{B^2}{dT^2} \right)$$

$$\frac{\mathrm{d}^2 L}{\mathrm{d}w \, \mathrm{d}\Delta a} = -\frac{T - B}{T^2} \frac{B}{\sqrt{d}}$$

$$\frac{\mathrm{d}^2 L}{\mathrm{d}\Delta a^2} = 0.$$

Under Assumption 5 from A.1, $T \to \infty$, $B$ stays negligible compared to $T$, and $d$ is large enough so that $\sqrt{d}$ is negligible in front of $d$, so that $\frac{T-1}{T^2}$ becomes $\frac{1}{T}$, $\mathrm{d}_w^2 L$ becomes $\frac{1}{dT}$ and it is therefore

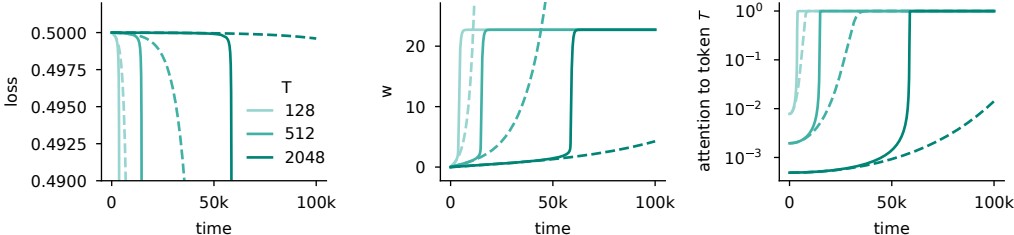

Figure 7: **Comparison of the early dynamics (plain lines) with their corresponding linear approximation around linear conditions (dashed lines).** $d$ is here fixed to $512$.

negligible in front of the cross-term second-order derivative. The gradient flow dynamics around initialization becomes

$$\begin{pmatrix} \dot{w} \\ \dot{\Delta a} \end{pmatrix} = \begin{pmatrix} \frac{B}{\sqrt{dT}} \\ 0 \end{pmatrix} + \begin{pmatrix} 0 & \frac{B}{\sqrt{dT}} \\ \frac{B}{\sqrt{dT}} & 0 \end{pmatrix} \begin{pmatrix} w \\ \Delta a \end{pmatrix} + o(\|w\| + \|\Delta a\|). \qquad (15)$$

Figure 7 provides a visualization of how well these approximate dynamics match the original one.

The Hessian matrix has a positive eigenvalue $\frac{B}{\sqrt{dT}}$ and one negative one $-\frac{B}{\sqrt{dT}}$. This implies that the initial parameters are in the vicinity of an unstable fixed point and that the negative eigenvalue of the Hessian will dictate how fast we are escaping these initial conditions. Its eigenvalues are

$$\lambda_\pm = \pm \frac{B}{\sqrt{dT}}$$

with eigenvectors

$$v_\pm = \frac{1}{\sqrt{2}} \begin{pmatrix} 1 \\ \pm 1 \end{pmatrix}.$$

Let $P$ be the change of basis matrix defined by these eigenvectors. This matrix $P$ transforms the original coordinates $(w, \Delta a)$ to the eigenbasis coordinates, $(z_+, z_-)$. The dynamics in the new basis is

$$\begin{pmatrix} \dot{z}_+ \\ \dot{z}_- \end{pmatrix} = \frac{B}{\sqrt{2dT}} \begin{pmatrix} 1 \\ 1 \end{pmatrix} + \frac{B}{\sqrt{dT}} \begin{pmatrix} 1 & 0 \\ 0 & -1 \end{pmatrix} \begin{pmatrix} z_+ \\ z_- \end{pmatrix}.$$

This yields

$$z_+ = \frac{1}{\sqrt{2}} \left( \exp(\lambda t) - 1 \right)$$

$$z_- = \frac{1}{\sqrt{2}} \left( 1 - \exp(-\lambda t) \right)$$

and

$$w = \sinh(\lambda t)$$
$$\Delta a = \cosh(\lambda t) - 1$$

**Initial evolution of the loss.** We can now derive a close-form equation for the temporal evolution of the quadratic approximation of the loss: as

$$L(w, \Delta a) = 0.5 - \lambda w - \lambda w \Delta a + o(\|w\|^2 + \|\Delta a\|^2),$$

we get that the loss is initially approximately equal to

$$L \approx \frac{1}{2} - \lambda \sinh(\lambda t) - \lambda \sinh(\lambda t)(\cosh(\lambda t) - 1)$$

$$= \frac{1}{2} - \lambda \sinh(\lambda t) \cosh(\lambda t)$$

$$= \frac{1}{2} \left( 1 - \lambda \sinh(2\lambda t) \right)$$

using the identity $\sinh(x)\cosh(x) = \frac{1}{2}\sinh(2x)$.

**Time needed to escape the initialization plateau.** We can now compute the time $T_\varepsilon$ it will take for the (approximate) loss to be equal to $(1 - \varepsilon)$ its initial value. From this definition, we get that $T_\varepsilon$ satisfies

$$\frac{1}{2}(1 - \lambda \sinh(2\lambda t)) = \frac{1 - \varepsilon}{2},$$

that is

$$T_\varepsilon = \frac{\sqrt{d}T}{2B} \operatorname{arcsinh}\left(\frac{\varepsilon\sqrt{d}T}{B}\right).$$

Given that we are in the regime of large $T$ and $d$, we finally get

$$T_\varepsilon = \frac{\sqrt{d}T}{2B} \log\left(\frac{2\varepsilon\sqrt{d}T}{B}\right)$$

by using the fact that $\operatorname{arcsinh}(x) \sim \frac{1}{2}\log(x)$ as $x \to \infty$.

### A.3.3 Phase transition and late phase dynamics

Once we escape initial conditions and attention to the relevant token(s) starts increasing, the impact of the other tokens on the loss starts becoming negligible, the loss approximately becomes

$$L \approx \frac{1}{2d}\left(B\alpha w - \sqrt{d}\right)^2. \tag{16}$$

This loss features multiplicative interactions akin to a one-hidden-layer neural network, and therefore it comes as no surprise that the loss exhibits similar sharp phase transitions as the ones observed when learning these networks, cf. Saxe et al. [29] for an in depth analysis of these dynamics (one needs to linearize the mapping $\Delta a \mapsto \alpha$ to get the exact mapping to this set of results).

The learning dynamics exhibits an interesting behavior after the phase transition: the projection of $w$ on $W^*$ starts decreasing, as seen on Figure 6 for example. In the following, we will argue that $w$ is close to equilibrium in that phase and that learning consists mainly in slowly pushing $\Delta a$ to infinity and that the corresponding equilibria decrease. We will show this by investigating the structure of the Hessian when $w$ is at equilibrium.

First, let us compute the value that $w$ takes at equilibrium, which requires solving the equation $\nabla_w L = 0$. Using Equation 13, it gives

$$\left(\frac{(1 - B\alpha)^2}{d(T - B)} + \frac{(B\alpha)^2}{d}\right)w = \frac{B\alpha}{\sqrt{d}},$$

that is

$$w^\infty(\alpha) := \left(\frac{(1 - B\alpha)^2}{d(T - B)} + \frac{(B\alpha)^2}{d}\right)^{-1} \frac{B\alpha}{\sqrt{d}}.$$

We plot $w^\infty$ as a function of $\alpha$ in Figure 8.left.

Let us now consider the case in which $B\alpha$ converges to 1, that is $B\alpha = 1 - \varepsilon$ with $\varepsilon \to 0$. This gives

$$w^\infty = \frac{B\alpha d}{(B\alpha)^2\sqrt{d}} + o(\varepsilon^2) = \frac{\sqrt{d}}{1 - \varepsilon} + o(\varepsilon^2).$$

We can then plug this value into the different components of the Hessian using the second derivatives we calculated for the simplified loss:

$$\frac{d^2 L}{dw^2} = \frac{1}{d} - \frac{2\varepsilon}{d} + o(\varepsilon^2)$$

$$\frac{d^2 L}{dw\, d\Delta a} = \frac{B\varepsilon}{\sqrt{d}} + o(\varepsilon^2)$$

$$\frac{d^2 L}{d\Delta a^2} = \left(\frac{B + B^2}{T - B} + B^2\right)\varepsilon + o(\varepsilon^2).$$

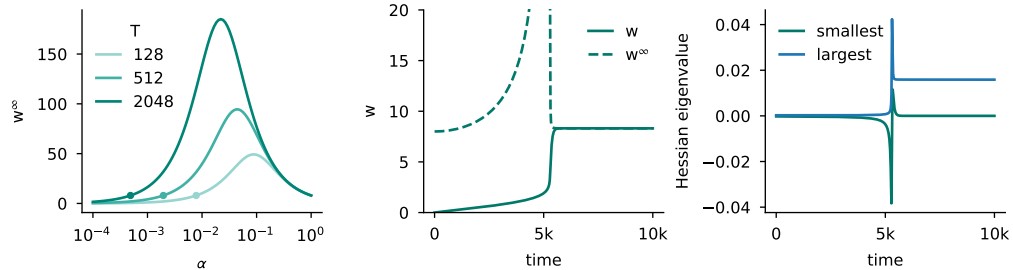

Figure 8: (left) $w^\infty$ as a function of $\alpha$ for different $T$ values ($d = 64$, $B = 1$). (middle) Evolution of $w$ on the gradient flow dynamics ($T = 512$, $d = 64$, $B = 1$) and comparison with its instantaneous equilibrium value $w^*$. (right) Evolution of the smallest and largest eigenvalue of the Hessian of the loss along the gradient flow dynamics.

The Hessian is thus equal to

$$
H(w^\infty, \Delta a) = \begin{pmatrix} \frac{1-2\varepsilon}{d} & \frac{B\varepsilon}{\sqrt{d}} \\ \frac{B\varepsilon}{\sqrt{d}} & \left(\frac{B+B^2}{T-B} + B^2\right)\varepsilon \end{pmatrix} + o(\varepsilon^2)
$$

when $w^\infty$ is at optimality and $B\alpha$ close to 1 (i.e. $\Delta a \to \infty$). As $\varepsilon$ goes to 0, the loss becomes increasingly flat in the $\Delta a$ dimension, highlighting that $\Delta a$ will be the bottleneck in terms of learning and $w$ will always be at its equilibrium values $w^\infty$. We confirm this in simulation in the middle panel of Figure 8.

It is now worth comparing the loss landscape structure in this late phase compared to the one early on during learning. In the early stages, the eigenvalues are small ($B/\sqrt{dT}$) and $w$ and $\Delta$ both contribute to them. At the end of learning, the loss is much sharper in the $w$ direction ($1/d$) than in the $\Delta a$ direction dimension ($\varepsilon$), because of the softmax within the attention. From this analysis of the two extremes of the learning dynamics, in discrete time, $w$ would be the bottleneck in terms of the learning rate, due to the final sharpness of the loss. A complete picture is hard to obtain analytically because calculations become more involved in the middle of learning, so we resort to simulations and plot the results in Figure 8.right. We find that the behavior we described analytically holds for reasonably long in the early and late dynamics. In the middle of the dynamics, the geometry of the loss landscape changes drastically and the loss is the sharpest at this time (and thus the maximum learning rate we can take in discrete time will be dependent on geometry of the loss around the phase transition).

### A.3.4 Empirical validation

In these sections, our aim is to verify whether our theory still captures the behavior of the model when the assumptions we made are not met. In particular, we will relax Assumption 3 ($W^*$ has unit norm columns), 4 ($W$ initialized to 0), 5 (long sequences) and partially Assumption 1 (gradient flow dynamics). Indeed, we sample $W$ and $W^*$ from $\mathcal{N}(0, 1/d)$, take $T$ to be 256 (thus finite), $d$ to be 256 and consider stochastic gradient descent dynamics with batch size 32 and learning rate 1. We report the comparison of the evolution of the loss, the attention $\alpha$ to the relevant to the $T$-th token, and the projection $w$ of the weights $W$ on $W^*$ in Figure 9 for both the reduced dynamics (Theory line, as in Equations 13 and 14) and for simulations (with the parameters described above). Overall, we find that the simplified dynamics is able to depict the ones of the actual model quite accurately.

### A.4 Analysis of the dynamics with cross-sample repetition

When considering cross-sample repetition, $\Sigma_T$ will have a larger magnitude in the direction of the token $\tilde{x}$ that appears more frequently during learning. As a result, weights learn faster in that direction and having a single scalar to capture the behavior of the entire weight matrix is no longer possible. In the following, we show how to reduce it to two scalars using similar techniques as before, as well as proceed to the analysis.

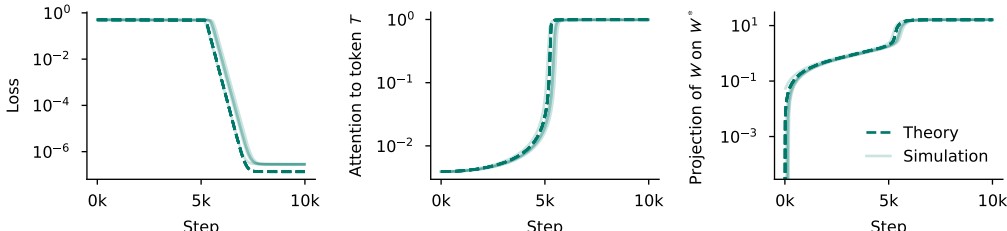

Figure 9: **The gradient flow dynamics on the simplified loss of Equation 12 match the stochastic gradient descent dynamics of the original objective.** We report the dynamics of 5 different seeds for the simulation lines. Changing the seed modified the target mapping, the sampling process, as well as the model initialization. Additional details are provided in Section A.3.4.

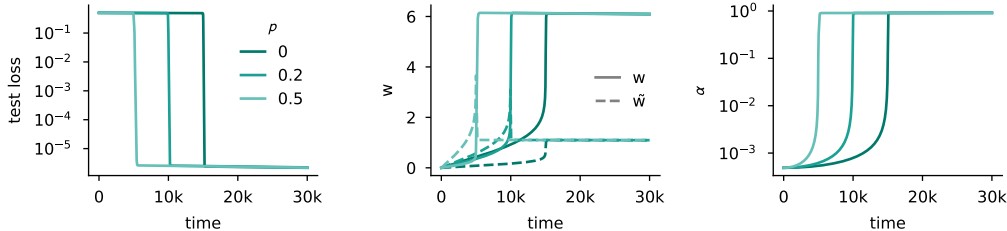

Figure 10: **Cross-sample repetition speeds up emergence.** This is because the repeated component $\tilde{w}$ (dashed lines in the middle plot) learns faster, which leads to an earlier increase of attention $\alpha$ to the relevant token (right) and faster learning overall (left). The results reported here were obtained by simulating the gradient flow on the loss of Equation 17 for $T = 512$ and $d = 64$.

### A.4.1 Reducing the learning dynamics to three dimensions

Cross-sample repetition does not affect attention directly, so we can still capture there entire behavior of attention with

$$\alpha = \frac{1}{(T-1)\exp(-\Delta a) + 1}.$$

Note that we do not consider any in-context repetition here for simplicity, but all our results can be extended to this case. Let us now look at the weights. Without loss of generality, we can assume $\tilde{x} = [1, 0, \cdots, 0]^\top$, which yields

$$\Sigma_T = \text{diag}\left(p + \frac{1-p}{d}, \frac{1-p}{d}, \cdots, \frac{1-p}{d}\right).$$

The gradient of the loss with respect to the $i$-th column $W_{:i}$ of the weight matrix therefore becomes

$$\nabla_{W_{:i}} L = \sum_{t > T-B} \alpha_t \left(p\delta_{i=1} + \frac{1-p}{d}\right)\left(\sum_{t > T-B} \alpha_t W_{:i} - W_{:i}^*\right) + \sum_{t \leq T-B} \frac{\alpha_t^2}{d} W_{:i}.$$

Importantly, when initialized at 0, each column of $W$ will evolve on the line spanned by the corresponding column of $W^*$. However, they will not move at the same speed as the first column has different dynamics than the other columns. We therefore introduce two scalars $\tilde{w}$ and $w$ such that $W_{:1} = \tilde{w}W_{:1}^*$ and $W_{:i} = \frac{w}{\sqrt{d-1}}W_{:i}^*$. As for the analysis in the previous section, the $\sqrt{d-1}$ scaling factor is here to ensure that the learning speed coincide. To keep the notation as concise as possible, we introduce $\tilde{\sigma} := p + \frac{1-p}{d}$ and $\sigma := \frac{1-p}{d}$.

Under these assumptions, the loss simplifies to

$$L = \frac{1}{2}\left(\frac{(1-\alpha)^2(w^2 + \tilde{w}^2)}{d(T-1)} + \tilde{\sigma}\left(\alpha\tilde{w} - 1\right)^2 + \sigma\left(\alpha w - \sqrt{d-1}\right)^2\right). \tag{17}$$

The gradients flow dynamics on this loss are reported on Figure 10. The gradients are

$$\nabla_{\tilde{w}} L = \frac{(1-\alpha)^2 \tilde{w}}{d(T-1)} + \alpha \tilde{\sigma}(\alpha \tilde{w} - 1) \tag{18}$$

$$\nabla_w L = \frac{(1-\alpha)^2 w}{d(T-1)} + \frac{\alpha(1-p)(\alpha w - \sqrt{d-1})}{d} \tag{19}$$

$$\nabla_{\Delta a} L = \alpha(1-\alpha) \left( -\frac{(1-\alpha)(w^2 + \tilde{w}^2)}{d(T-1)} + \tilde{w}\tilde{\sigma}(\alpha \tilde{w} - 1) \right. \tag{20}$$

$$\left. + w\sigma(\alpha w - \sqrt{d-1}) \right). \tag{21}$$

The entries of the Hessian are

$$\frac{\mathrm{d}^2 L}{\mathrm{d}\tilde{w}^2} = \frac{(1-\alpha)^2}{d(T-1)} + \alpha^2 \tilde{\sigma}$$

$$\frac{\mathrm{d}^2 L}{\mathrm{d}w^2} = \frac{(1-\alpha)^2}{d(T-1)} + \alpha^2 \sigma$$

$$\frac{\mathrm{d}^2 L}{\mathrm{d}w\,\mathrm{d}\tilde{w}} = 0$$

$$\frac{\mathrm{d}^2 L}{\mathrm{d}\tilde{w}\,\mathrm{d}\Delta a} = \alpha(1-\alpha) \left( -\frac{(1-\alpha)\tilde{w}}{d(T-1)} + (2\alpha\tilde{w} - 1)\,\tilde{\sigma} \right)$$

$$\frac{\mathrm{d}^2 L}{\mathrm{d}w\,\mathrm{d}\Delta a} = \alpha(1-\alpha) \left( -\frac{(1-\alpha)w}{d(T-1)} + \left(2\alpha w - \sqrt{d-1}\right)\sigma \right)$$

$$\frac{\mathrm{d}^2 L}{\mathrm{d}\Delta a^2} = \frac{1-2\alpha}{\alpha(1-\alpha)} \nabla_{\Delta a} L + \alpha(1-\alpha) \left( \frac{(w^2 + \tilde{w}^2)}{d(T-1)} + (\tilde{\sigma}\tilde{w}^2 + \sigma w^2) \right)$$

### A.4.2 Approximate dynamics around initialization

Around initialization we get

$$\frac{\mathrm{d}}{\mathrm{d}t} \begin{pmatrix} \tilde{w} \\ w \\ \Delta a \end{pmatrix} = \begin{pmatrix} \frac{\tilde{\sigma}}{T} \\ \frac{\sigma\sqrt{d-1}}{T} \\ 0 \end{pmatrix} + \begin{pmatrix} -\frac{T-1}{dT^2} - \frac{\tilde{\sigma}}{T^2} & 0 & \frac{\tilde{\sigma}(T-1)}{T^2} \\ 0 & -\frac{T-1}{dT^2} - \frac{\sigma}{T^2} & \frac{\sigma(T-1)\sqrt{d-1}}{T^2} \\ \frac{\tilde{\sigma}(T-1)}{T^2} & \frac{\sigma(T-1)\sqrt{d-1}}{T^2} & 0 \end{pmatrix} \begin{pmatrix} \tilde{w} \\ w \\ \Delta a \end{pmatrix}$$

Under Assumption 5, this dynamics becomes

$$\frac{\mathrm{d}}{\mathrm{d}t} \begin{pmatrix} \tilde{w} \\ w \\ \Delta a \end{pmatrix} = \begin{pmatrix} \frac{p}{T} \\ \frac{1-p}{\sqrt{d}T} \\ 0 \end{pmatrix} + \begin{pmatrix} 0 & 0 & \frac{p}{T} \\ 0 & 0 & \frac{1-p}{\sqrt{d}T} \\ \frac{p}{T} & \frac{1-p}{\sqrt{d}T} & 0 \end{pmatrix} \begin{pmatrix} \tilde{w} \\ w \\ \Delta a \end{pmatrix}.$$

The characteristic polynomial of the Hessian is

$$X^3 - X \left( \frac{(1-p)^2}{dT^2} + \frac{p^2}{T^2} \right)$$

which means that the eigenvalues of the Hessian are 0 and

$$\pm \frac{1}{\sqrt{d}T} \sqrt{p^2 d + (1-p)^2},$$

which provides a scaling factor for the exit time of

$$\frac{\sqrt{d}T}{\sqrt{p^2 d + (1-p)^2}}.$$

As a sanity check, we can remark that when there is no cross-sample repetition, we recover the same scaling factor as in our previous analysis. As $p$ increases, it progressively removes the effect of the dimension $d$ in the escape time.

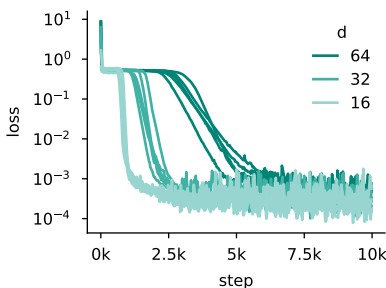

Figure 11: **The learning dynamics of Transformers exhibit sharp phase transitions in the single-location task.** Here, $T = 128$ and the architecture and training details are the one described in Appendix D.3.

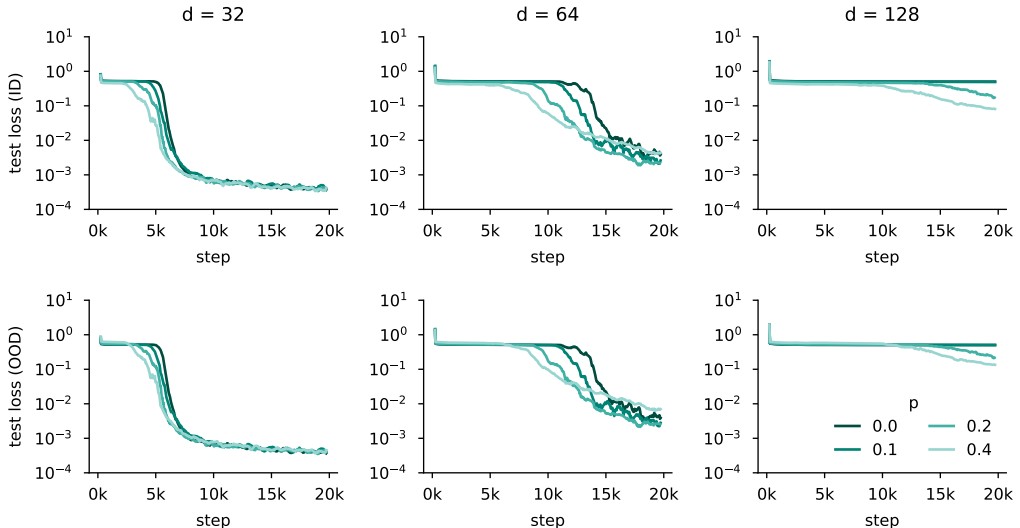

Figure 12: **Effects of cross-sample repetition on the learning dynamics of a Transformer in the linear regression task.** The top line corresponds to the test loss on the same data distribution than the one on which the network is trained, that is with repetition. The bottom line is data without any repetition. The results are obtained for $T = 256$. See Appendix D.3 for the rest of the training details.

## B   Details and additional analysis for the linear regression task

### B.1   Implementation of the task

For our experiments, we implement a slightly different version of the task than the one we introduced in the main text and that we used for our theoretical analysis. The difference lies in where the relevant token is and whether there exists a feature to indicate it:

–  **Random relevant token position**. In the simple version of the task, the relevant token was always presented at position $T$ for convenience, as the output of the toy attention model we consider does not depend on the specific position. For experiments with more realistic Transformer models, we make this position random to avoid skip connections playing a specific role, and we resample it for every new sequence. The only exception to this is the task specifics ablation of Figure 14, in which we sometimes fixed it throughout the entire training run (the relevant position being still randomly sampled at the beginning of training).

–  **Relevant token feature**.  As the position of the relevant token is resampled for every new sequence, we need to provide information to the network about which token is relevant for the

task. To this end, we append an extra feature to each input token that is 1 whenever the token is relevant. As in the previous point, the only experiment in which this feature is not added to the input is the ablation of Figure 14.

### B.2 Examples of learning curves

Figures 11 and 12 provide examples of the learning dynamics of Transformers on the variation of the single-location linear regression task described in the previous section.

### B.3 Additional ablations on the impact of task variation, model size and optimizer on plateau length

In the main text, we claim that the architecture, the details of the task, as well as the optimizer change the dependence of the plateau length on the different data parameters (here we study $d$ and $T$). These changes collectively explain why the theoretical exponents of the power laws we obtained in our simplified regime do not directly translate to practice. This section provides the ablation underlying this claim; we detail them by increasing importance.

– **Architecture.** The toy model we consider has a single head and a single layer whereas the Transformer architecture we consider has 2 layers and 4 heads. In Figure 13, we ablate the number of layers and number of heads. We find that they have some importance, by slowing the learning process (note that we did not tune the learning rate to each architecture size). However, they do not significantly alter the dependency in $d$ and $T$ captured through the power law exponents.

– **Task specifics.** In the more realistic version of the task, the Transformer is provided with an extra input feature which indicates whether the token is relevant for the task, and the relevant token position is random. However, the toy model does not have access to this feature (it cannot use it) and the position is fixed. In Figure 14, we ablate these different choices and find that randomizing the position significantly increase the dependence in $T$ and, to a smaller extent, the dependence on $d$. Interestingly, the power laws are almost the same for fixed position when the relevant feature is given to the model or not. We argue that this may be because the relevant feature provides a weaker statistical signal, likely because of the initial embedding layer from dimension $d$ to 256, than the positional encoding. Learning to use would therefore be slower, and the network eventually ignores this feature.

– **Optimizer.** Our theoretical were obtained by analyzing the gradient flow dynamics. While gradient flow has tight links with gradient descent, and thus the emergence time under the gradient flow is approximately proportional to the number of steps before emergence (assuming small enough learning rates), this does not hold for Adam. To test how much Adam changes emergence time, we ablate the choice of the optimizer in Figure 15. We find that Adam to be significantly faster than SGD (up to almost two order of magnitudes for the hard versions of the task), both in absolute terms and in sensitivity to the difficulty of the task.

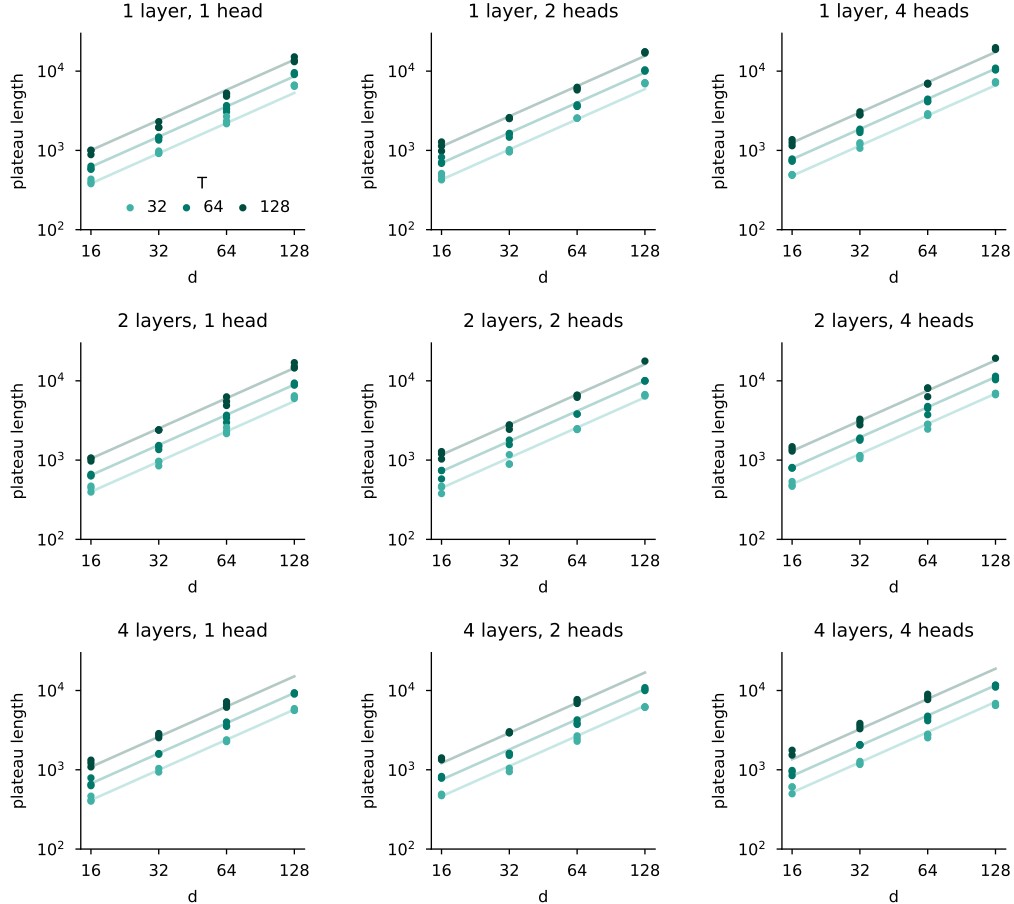

Figure 13: **Changing the width of the network (by increasing the number of heads) increases time spent on the initial plateau more significantly than increasing the depth of the network.** We plot the evolution of the plateau length, for Transformers with different number of layers and number of heads. We obtain the following scaling law: $T_{\mathrm{plateau}} = 1.03\, d^{1.27}\, T^{0.70}\, N_{\mathrm{layers}}^{0.06}\, N_{\mathrm{heads}}^{0.16}$ ($R^2 = 0.995$), which corresponds to the lines in the plots above.

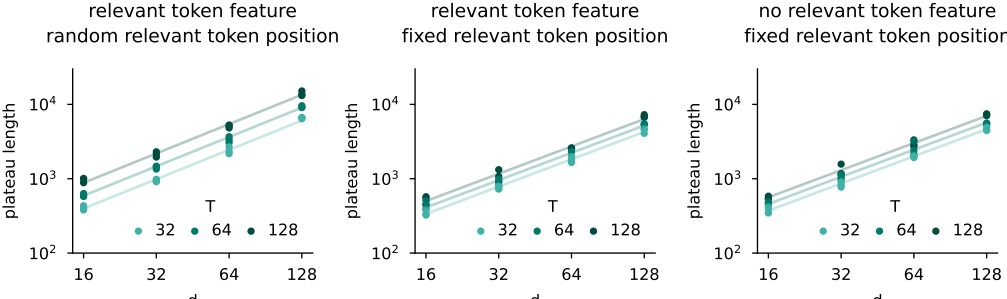

Figure 14: **Emergence is faster when positional information is available than when the network has to use semantic information.** In this set of experiments, we vary the nature of the task. The network can be given a "relevant token feature" indicating whether the token is relevant for the task is given (that is semantic information about the nature of the token). The position of the relevant token for the task can be either fixed or random. If it is random, the network must use semantic information to solve the task. If not, it (may) use positional information. In particular, this means that the network cannot solve the task when it has no access to the relevant token feature and the positions are random; this is why we do not report results in this configuration here. The different scaling laws plotted are:

(*left*) $T_{\mathrm{plateau}} = 1.43 \, d^{1.31} \, T^{0.58}$, $R^2 = 0.998$ (relevant feature, random position).
(*middle*) $T_{\mathrm{plateau}} = 4.20 \, d^{1.22} \, T^{0.29}$, $R^2 = 0.996$ (relevant feature, fixed position).
(*right*) $T_{\mathrm{plateau}} = 4.61 \, d^{1.21} \, T^{0.30}$, $R^2 = 0.999$ (no relevant feature, fixed position).

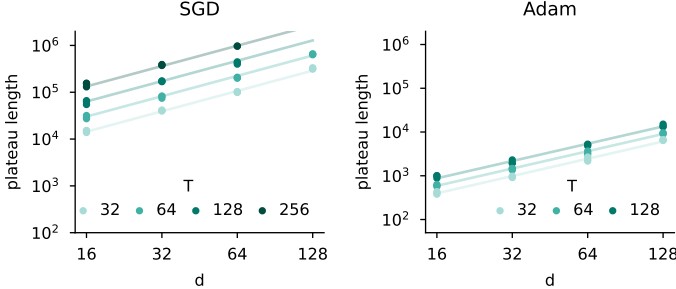

Figure 15: **Switching SGD for Adam leads to an important acceleration of learning.** Here, we train a single head single layer transformer with (left) SGD with a learning rate of $5 \cdot 10^{-3}$ and (right) Adam with a learning rate of $10^{-4}$ (these learning rates are manually tuned). Changing the optimizer has a huge effect on emergence time, both in terms of absolute time but also of scaling as the problem gets harder: $T_{\mathrm{plateau}} = 6.42 \, d^{1.44} \, T^{1.07}$ ($R^2 = 0.997$) for SGD and $T_{\mathrm{plateau}} = 1.45 \, d^{1.31} \, T^{0.57}$ for Adam ($R^2 = 0.998$).

# C  Details and additional experiments for the in-context associative recall task

## C.1  Implementation of the task

Recall that each sequence consists of $N_{\text{pairs}}$ key-value token pairs (5 such pairs in the example below) followed by a query token (Z below), as shown in the following example:

$$Y\ I \quad A\ X \quad U\ R \quad Z\ Y \quad C\ A \quad Z\ ?$$

The number of possible tokens is the vocabulary size $N_{\text{tokens}}$ and each of these tokens has a corresponding one-hot encoding. Both keys and values are sampled from the same pool of tokens.

The generative process behind the task is as follows:

- **Set up query distribution.** We define a query distribution $p_{\text{query}}$ which corresponds to uniformly sampling one of the 2-repeated tokens with probability $p$ and uniformly sampling all the tokens with probability $1 - p$ (recall that $p$ is the cross-sample repetition probability). Said otherwise, this probability distribution has mass $\frac{p}{2} + \frac{1-p}{N_{\text{tokens}}}$ on the repeated tokens and mass $\frac{1-p}{N_{\text{tokens}}}$ on the rest of the tokens.

- **Sample the in-context mapping.** For each new sequence, we sample the mapping between keys and queries. This mapping is injective, meaning that two different keys will always be associated with different values. In practice, we sample a random permutation for this mapping.

- **Sample the query**. For each new sequence, we sample the query following the probability distribution $p_{\text{query}}$ defined when creating the task. The target output for the task, will be the value indicated by the in-context mapping.

- **Fill the context.** For each new sequence, we finally fill the context as follows. We first decide whether we put the query (and its corresponding value) in the $N_{\text{pairs}}$ possible positions by sampling a Bernoulli variable with success probability $\frac{B}{N}$ (on expectation the query thus appears $B$ times in the context). We then fill the rest of the context by sampling keys from the $N_{\text{tokens}} - 1$ remaining tokens without replacement, and appending their corresponding values directly after.

## C.2  Learning dynamics and additional analysis

We here provide two additional results: the learning dynamics on the training data, which show that repetition accelerates emergence in similar ways to what is predicted in our theory (Figure 16) and the evolution of the attention scores over the course of learning, which confirms that emergence occurs jointly with attention to relevant tokens getting sparser (Figure 17). We can make additional comments regarding the last point: it is interesting to see that all heads in the same layer have similar attention to relevant tokens, and that the copying mechanism (to group together the key and the value) only occurs in layer 3 and the selection mechanism only in layer 4.

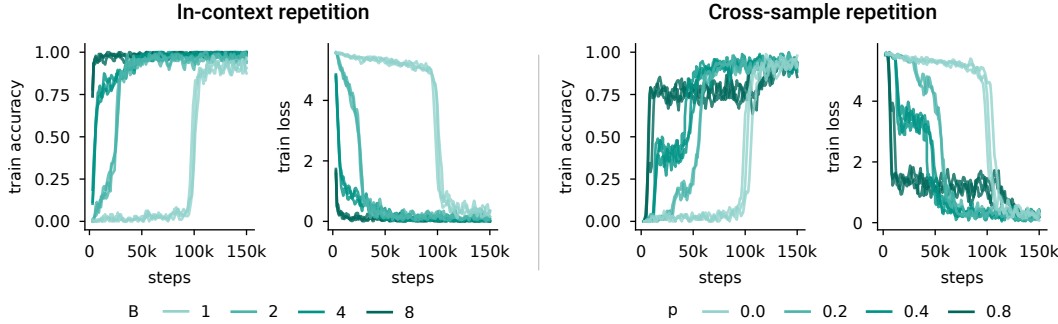

Figure 16: **Repetition accelerates emergence in the associative recall task as per the theory.** This plot is the training counterpart of the plots of Figure 5. As we measure the performance on the training data, there is no overfitting and repetition only speeds up emergence.

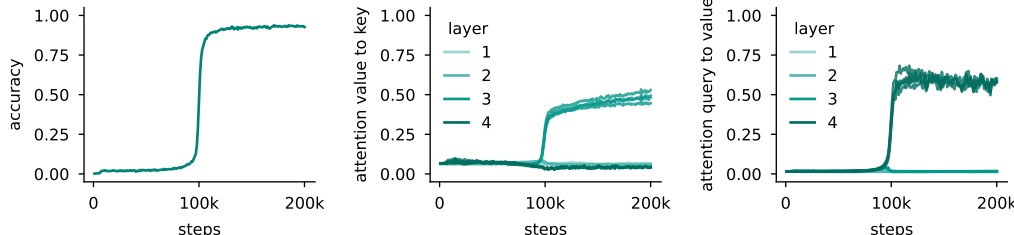

Figure 17: **The emergence of associative recall co-occurs with an increase of attention to relevant tokens.** (left) Evolution of the performance of the model over time, as in Figure 4 left ($N_{\mathrm{pairs}} = 32$ and $N_{\mathrm{tokens}} = 256$). There is no repetition in this experiment. (middle) Evolution of the attention from the relevant value token to the relevant key token (the first layer of a two-layers manually constructed attention head would have this attention value equal to 1). (right) Evolution of the attention from the query token to the relevant value token (the second layer of the induction head would get this value equal to 1).

# D  Methods

## D.1  Measurement of the plateau length

To quantify the emergence time of sparse attention, we measure the duration of the initial plateau in the learning dynamics. We define the plateau length as the number of training steps required for the model to reach a specific threshold performance from initialization. This performance threshold depends on the task we consider:

– For the linear regression task with the toy model, we define the plateau length as the time required for the test loss to decrease to $0.2$ of its initial value when following the gradient flow dynamics, that is $0.1$. The test loss is the loss without any form of repetition for the vanilla dynamics (Figure 1.a) or for the dynamics with cross-sample repetition (Figure 2 right), and with the same $B$ value as during training for in-context repetition. Note that we use a test loss with repetition for the latter as the toy model cannot generalize to the no repetition case.

– For the linear regression task learned with a Transformer, we use the same threshold as above, except for the cross-sample repetition where we used a threshold of $0.4$. This threshold being different partially explains why the lighter lines in Figure 3 middle and right do not map (the other reason being that we additionally tuned the learning rate for the right plot). The reason behind this change comes from the shape of the learning curves: after emergence, the learning speed can be drastically different for different $p$ values (see Figure 12, and hence we need it to be closer to $0.5$, the initial loss value, to accurately capture the emergence time.

– For the in-context associative recall task, we pick an accuracy threshold, here $5\%$, as the random guess loss varies as a function of the vocabulary size.

It is important to note that in the first case, the plateau length corresponds to a physical time, whereas for the last two cases it corresponds to a number of optimization steps.

## D.2  Fitting power laws on plateau length

To quantify the relationship between plateau length and various factors (sequence length $T$, input dimension $d$, in-context repetition $B$, cross-sample repetition probability $p$), we fit power laws to our empirical measurements using least squares regression on log-transformed data. We also fit power laws on the number of heads and the number of layers in 13 according to similar principles to what is detailed below.

For the general form of our scaling laws, we assume:

$$T_{\text{plateau}} = C\, d^{\alpha} \left(\frac{T}{B}\right)^{\beta} f(p, d)^{\gamma} \tag{22}$$

where $C$ is a constant, $\alpha$, $\beta$, and $\gamma$ are the scaling exponents, and $f(p, d) = \sqrt{p^2 d + (1 - p)^2}$ is a function of the repetition probability (which is rooted in our theoretical analysis of Appendix A.4. Note that when considering the associative recall task, the vocabulary size is replacing $d$ and the number of pairs is replacing $T$.

We perform the fitting by linearizing this relationship through a logarithmic transformation:

$$\log T_{\text{plateau}} = \log C + \alpha \log d + \beta \log\left(\frac{T}{B}\right) + \gamma \log f(p, d) \tag{23}$$

and performing a linear regression to find the missing coefficients. We report the score ($R^2$) of the linear fit as a measure of fit quality for all scaling laws. Most reported scaling laws had $R^2 > 0.98$, indicating excellent fit to the empirical data. For the cross-sample repetition results of Figure 3 right, where we could not fit accurately any power law on the obtained results.

More precisely, the exact parameters we fit for each plot are:

– Figure 1.c: $C$, $\alpha$ and $\beta$ ($B$ is fixed to $1$).
– Figure 2 left: $C$, $\alpha$ and $\beta$ ($T$ is fixed to $4096$).
– Figure 2 right: $C$, $\alpha$, $\beta$ and $\gamma$, such that $2\alpha = \beta = \gamma$ ($T$ is fixed to $4096$).

– Figure 3 left and middle: $C$, $\alpha$ and $\beta$. The same power law is fitted on the data of the two plots.
– Figure 4 right: $C$, $\alpha$ and $\beta$ ($B$ is fixed to 1).

## D.3 Architectural and training details

We report the default hyperparameters we use in our experiments in Table 1 and report the deviations we make to this setup below:

– In Figure 3 right, we additionally tune the learning rate (in $\{10^{-4}, 3 \cdot 10^{-4}\}$) based on the shortest average plateau length as we found cross-sample repetition to significantly change the learning rate the Transformers we considered need.
– In Figure 4 right, we additionally tune the learning rate (in $\{10^{-5}, 3 \cdot 10^{-5}, 10^{-4}\}$) as large vocabulary sizes and number of pairs need smaller learning rates.
– In Figure 13, we ablate the number of layers and the number of heads in the Transformer architecture.
– In Figure 15, we ablate the choice of Adam as optimizer and replace it by stochastic gradient descent. Note that we here consider a single layer and a single head.

It should be additionally noted that we adapt the number of iterations depending on the difficulty of the task and roughly aim to end training significantly after emergence. However, for the most challenging versions of the task (e.g., large $T$ values in Figure 3), the phase transition sometimes occurs after the maximum number of training steps we attribute to the task (in that case 50k steps).

| Parameter Type | Theory | Linear regression | Associative recall |
|---|---|---|---|
| **Architecture Parameters** | | | |
| Model | Toy model | Transformer | Transformer |
| Layers | 1 | 2 | 4 |
| Number of heads | N/A | 4 | 4 |
| Embedding dimension | N/A | 256 | 256 |
| QK dimension | N/A | 64 | 64 |
| MLP factor | N/A | 4 | 4 |
| Positional encoding | N/A | sinusoidal | sinusoidal |
| Layer normalization | No | No | No |
| Skip connections | No | Yes | Yes |
| MLP between layers | No | Yes | Yes |
| **Training Parameters** | | | |
| Optimizer | GD | Adam | Adam |
| Learning rate | 1.0 | $10^{-4}$ | $10^{-4}$ |
| Scheduler | None | None | None |
| Batch size | Full | 32 | 32 |
| Seeds | N/A | 5 | 3 |

Table 1: Default hyperparameters for the different types of experiments.

## D.4 Reproducibility

Our theoretical simulations were run in JAX [68] and our Transformer experiments in Py-Torch [69]. All our experiments were run on Nvidia RTX 3090 GPUs. The typical training run takes one hour, although training on shorter sequences can be significantly shorter. The code is publicly available at the following url: `https://github.com/NicolasZucchet/The-emergence-of-sparse-attention`.

