# OpenReview forum: "The emergence of sparse attention: impact of data distribution and benefits of repetition"
_NeurIPS.cc/2025/Conference — NeurIPS 2025 oral_

### Official Review · Reviewer_oc9H · 2025-06-24

**Clarity:** 2
**Significance:** 2
**Originality:** 3
**Rating:** 4
**Confidence:** 3

**Summary:**

The paper investigates the emergence of sparse attention patterns in Transformer models during training, focusing on how data distribution and repetition influence this process. The authors combine theoretical analysis with empirical experiments to uncover the mechanisms behind sparse attention emergence and its impact on model performance.

**Questions:**

Please see Weaknesses. My primary focus lies in points 1 through 5.

**Ethical Concerns:**

["NO or VERY MINOR ethics concerns only"]

**Final Justification:**

The authors theoretically and experimentally elucidate the advantages of in-context repetition and cross-sample repetition, which provides guidance for practical applications of large language models, such as training data allocation. However, a limitation lies in the need for further examination of the gap between simplified models and real-world applications.

**Limitations:**

yes, some Limitations are discussed in Sec. Discussion.

**Paper Formatting Concerns:**

There is no major formatting issue in this paper.

**Quality:**

2

**Strengths And Weaknesses:**

> **Strengths:**
>
> (1) Good visualizations.
>
> (2) They explore sparse attention as a lens to understand emergence during training.

> **Weaknesses:**
>
> (1) Theoretical analysis primarily focuses on the single-location linear regression task, which is insufficient for analyzing the training dynamics of models.
>
> (2) On line 87, it states 'the relevant token location is always the same.' This will lead to limited diversity in training tasks, creating a gap with large language models. Additionally, it will also restrict the scope of applicability for theoretical analysis.
>
> (3) Line 104 "The model’s parameters (a, W) are learned by minimizing the expected mean square error between predictions y and targets y*."  Lines 142-144 "attention layer allocates B times more weight to the relevant tokens compared to the case without any repetition".  So, in both scenarios, does the pair (w, a) end up learning the same thing in the end? Does "B times" imply that the value of 'a' can be different?
>
> (4) The authors need to more clearly present the corresponding evidence to support the claim made on Lines 185-187.
>
> (5) Lines 278-288, what are the differences and connections between "grokking" and "emergence"?

---

> ### Author Rebuttal · Authors · 2025-07-30
>
> We thank the reviewer for their feedback and for clearly expressing their concerns about our paper. We provide some clarification below, which we believe address all the concerns raised by the reviewer:
> 1. **Weakness: The single-location task is insufficient to analyze the training dynamics of real models.** The reviewer is right that our analysis (in particular the theoretical one) does not fully capture all aspects of the learning dynamics of large language models, trained on more sophisticated tasks. However, we would like to nuance this point. Our analysis was designed with minimality as a main goal: we wanted to find a **minimal setting** in which emergence occurs during training and in which we could see the benefits of repetition. This allows us to identify, for the first time, a simple mechanism (sparse attention) that is easier to understand intuitively, while being practically relevant as it is ubiquitous in trained Transformers, and has close links with induction heads in in-context learning. By design, we are not trying to quantitatively capture realistic learning dynamics, but rather get an **intuitive understanding of how different variables** (here sequence length, data dimension, or repetition) **affect learning**. We believe our analysis **provides insights about some non-trivial behaviors** that this simplification makes much easier to analyze and conceptually easier to grasp. And we hope that, in future work, these findings can be applied to wider settings to gain additional understanding.
> 2. **Weakness: Fixing the location of the relevant token limits the applicability of the results.** The location is indeed fixed in all experiments provided in Section 2. The ablations in Figure 14 show that randomizing the token position (and providing a feature to the Transformer that allows it to select this position) has **some effect** on the exact scaling laws that the plateau length follows, **but the overall finding remains unchanged**.
> 3. **Question: Values taken by $(w, a)$ depending on the B value?** $w$ always takes the same final value ($1$) whatever the B value. When there is in-context repetition, the attention is equally shared between all relevant tokens, so we have the identity $\alpha B = 1$ at the end of training with $\alpha$ being the attention given to one relevant token. This implies that the attention to each of the relevant tokens decreases as $B$ increases. In terms of $a$, the difference between the $a$ for relevant tokens and the $a$ for non-relevant tokens goes to infinity, whatever the $B$ value. To summarize, **increasing $B$ does not change the final function learned by the network, but only convergence speed.**
> 4. **Question: evidence for the claims of l.185-187?** We thank the reviewer for pointing out this issue. These sentences **summarize the findings of the section**, and the rest of the section provides evidence for these claims. We have made that clearer in the current version of the manuscript.
> 5. **Question: Differences and connections between grokking and emergence?** Emergence is a broad concept that goes beyond emergence during training (the type we study). **One can understand grokking as a specific form of emergence during training**, in which the generalization abilities of the model suddenly improve much later than the ability of the model to solve the training task. However, there are other forms of emergence during training, for example, there are cases in which emergence already occurs looking at the training loss only (we study one such example here).
>
> We hope that this answers the concerns raised by the reviewer, and remain available for additional clarifications. Given these answers, we would appreciate it if the reviewer could reevaluate our work.

---

> > ### Comment · Reviewer_oc9H · 2025-08-02
> > **Thank you for your response**
> >
> > I appreciate the author's efforts to address my concerns, and I have decided to adjust the score I previously assigned. However, additional algorithmic/strategic designs in the practical deployment of large language models would be beneficial, particularly regarding pre-training data allocation ratios.

---

### Official Review · Reviewer_qdKB · 2025-07-01

**Clarity:** 4
**Significance:** 4
**Originality:** 4
**Rating:** 6
**Confidence:** 5

**Summary:**

The paper considers the emergence of sparse attention. They adopt three settings: single-location linear regression, in-context repetition, and cross-sample repetition. In the simplified model, they discover a power law of the plateau length corresponds to the sequence length T and hidden dimension d. They build a simplified theoretical framework to explain the dynamics of attention weight a and value matrix W, matching the observation. They further use the theory to explain the benefits of in-context repetition and cross-sample repetition. They also extend the results to realistic transformers and learning problems.

**Questions:**

1. What is the motivation of removing layer norm in the 2-layer 4-heads Transformer?

2. Just curious, have you observed the linear growth of the norm of $W$ (the value matrix in attention layer) along the training dynamics in 2-layer 4-heads TF if having layer norm and using Adam?

**Ethical Concerns:**

["NO or VERY MINOR ethics concerns only"]

**Final Justification:**

The paper is both theoretically solid and interesting in practice. I do not see any major issue. Therefore, I decide to give strong accept.

**Limitations:**

See my previous comment.

**Quality:**

4

**Strengths And Weaknesses:**

Although the plateau is repeatedly discovered in in-context learning literature, the plateau length is rarely discussed. The authors make very interesting findings on the power law of the plateau length. They also make novel simplification of the training dynamics and provide theoretical insights on the observed power law. The insights from analyzing in-context repetition and cross-sample repetition are also interesting.

For the weakness, my major concern is the limited implication of the benefits of in-context repetition and cross-sample repetition. I understand that these are interesting findings. But they are restricted in the given in-context learning setup.

1. The more interesting direction of in-context repetition (bustiness) is its relation to in-weight learning. It is less surprising that more in-context repetition leads to shorter plateau length.
2. I would imagine that the cross-sample repetition is only useful given the special design of the in-context learning task. The benefits of data repetition in training, or the longer training (grokking), seem to come from reasoning mechanism (for example, see Nanda 2023), but barely relates to in-context learning. The coding example given in the last paragraph of the paper seems to be more related to reasoning rather than in-context learning. I also find it hard to relate the discoveries in the paper to active learning. In general, I feel the discussion in section 4 should be more carefully phrased.

Nanda, Neel, Lawrence Chan, Tom Lieberum, Jess Smith, and Jacob Steinhardt. 2023. “Progress Measures for Grokking via Mechanistic Interpretability.” ArXiv.org. 2023. https://arxiv.org/abs/2301.05217.

‌

---

> ### Author Rebuttal · Authors · 2025-07-30
>
> We thank the reviewer for their appraisal of our paper!
>
> We agree with the reviewer that the role of in-context repetition is rather intuitive.
> Please see our answer to BBS6 regarding the second weakness the reviewer mentions (specificity of our results to the in-context learning task). Given the generality of the mechanism we highlight, we expect cross-sample repetition to be useful beyond the in-context learning task we study; Zucchet et al. 2025 provide one such example.
>
> The link with active learning is as follows: we show that varying diversity might be a powerful way to speed up emergence, but can be damaging for generalization. This insight provides a **simple active learning algorithm**: whenever an agent sees that it is **stuck on a task**, it **decreases data diversity** (which it can, as an active learner), and it increases it afterwards when learning finally kicks off.
>
> We have made these points clearer in the current version of the manuscript.
>
> Regarding the questions asked by the reviewer:
> 1. **Question: why remove the layer norm?** Note that we only remove it for the single-location linear regression experiments, and keep it for the in-context learning experiments. Our main hypothesis is that it is difficult to always output 0s at specific locations with layer norm, but this is possible without. As a consequence, the layers have to implement the selection mechanism together, a cooperation that may take longer to learn.
> 2. **Question: linear growth of the value weights?** We haven’t experimented with this and do not have a good argument for why this would happen, as it would not help to improve the selection mechanism nor the required linear transformation of token embeddings. That said, we are happy to run that experiment if the reviewer has a hypothesis for why this would occur.

---

### Official Review · Reviewer_AeaF · 2025-07-01

**Clarity:** 3
**Significance:** 3
**Originality:** 3
**Rating:** 5
**Confidence:** 4

**Summary:**

This paper studies training time emergence (sudden jump in performance during training after an extended plateau) for a simplified in-context learning (ICL) task involving single-location linear regression. The authors simplify the problem to coupled dynamics of 2 scalar variables corresponding to learning the attention map and linear projection. They further study the effect of repetition (both in-context and cross-sample) to analyze how this affects emergence during training. Results in the toy model setup are verified qualitatively in Transformer training on an in-context recall task.

**Questions:**

1. I couldn't follow why “in-context repetition makes the attention pattern to be learned less sparse, thereby simplifying the task” (line 138) - could the authors clarify how making the pattern less sparse simplifies the task? (I understood how having in-context repetition "allocates B times more weight to the relevant tokens" in Line 143, but not how that simplifies the task itself)

2. Just to clarify, is Figure 1 (right) and 2 numerical simulation for the toy model? If so, is the takeaway from this plot that linearizing early-time dynamics is OK for estimating plateau length?

3. Line 117-118: “in these two equations, the first term is the relevant signal and the second term is the noise coming from non-relevant tokens.” - I couldn't follow this argument, could the authors clarify?

4. Line 189-190: “Removing layer normalization, which made solving the task more challenging” - is there a reason / hypothesis why this is true for this specific setup?

5. In-context associative recall task: Does induction head formulation map in any way to the toy model discussed in Eq (2)?

**Ethical Concerns:**

["NO or VERY MINOR ethics concerns only"]

**Final Justification:**

The authors have responded to my (clarifying) questions, and I've maintained my score of 5 for this submission.

**Limitations:**

Yes

**Quality:**

3

**Strengths And Weaknesses:**

Strengths

- Paper studies relevant problem of training-time emergence connecting it to attention map sparsity, and concretely shows how the two are related.
- Toy setup is well-formulated and somewhat represents realistic Transformer setups.
- Paper is overall well written and easy to follow.

Weaknesses

- Not a major weakness, but the theory setup of single-location in-context regression might be a bit too simplistic for many tasks (even the experiments considered in the paper). While it does seem to predict some emergent phenomena quite well, overall I am not sure if just a linear layer with a softmax-vector as Attention layer would be able to capture a lot of nuances in realistic setups.

---

> ### Author Rebuttal · Authors · 2025-07-30
>
> We appreciate the reviewer’s positive feedback on our work. We agree with the weakness they raised (see our answer to reviewer oc9h for nuance on it), and answer their questions below:
> 1. **Question: why does in-context repetition make the task easier?** With in-context repetition, the information relevant for the prediction is repeated multiple times within the context, and is thus **more correlated with the token to predict**. This provides a stronger statistical signal to the model, and thus speeds up learning. An alternative way of seeing that is that in-context repetition **reduces the effective sequence length** (it’s mathematically similar to having a sequence of length $T/B$ with only one relevant token). Given that learning time scales with sequence length, this effectively makes the task easier and speeds up learning.
> 2. **Clarification: takeaway from Figure 1 right and 2.** Figure 1 right is the simulation of the simplified dynamics (Eq 3 and 4), which comes at the price of a few assumptions and on which we derive the theoretical exit time. Figure 2 consists of the learning dynamics of the model of Equation 2 (without the assumptions). The takeaway from these figures is indeed that linearizing the early dynamics of the simplified model captures the plateau length in the model of Eq. 2.
> 3. **Clarification: signal and noise terms in equations 3 and 4.** At a high level, there are **two types of tokens** in this task, the relevant ones (**signal**) and the non-relevant ones (**noise**). The derivation of Equations 3 and 4, which can be found in Appendix A.2 and A.3, uses this insight to group the contributions to the loss of the tokens together, and Eq. 3 and 4 reflect this split.
> 4. **Question: hypothesis for why layer normalization makes the task harder?** We have not thoroughly investigated this question as this is likely specific to the single-location regression task (we use layer norms in the in-context learning experiments). Our main hypothesis is that it is difficult to always output 0s at a specific location with layer norm, but this is possible without. As a consequence, the layers have to implement the selection mechanism together, a cooperation that may take longer to learn.
> 5. **Question: how does the induction head map to equation 2?** We would first like to clarify that equation 2 is the equation of our toy model and that **it can not directly implement an induction head** (it is not a sequence to sequence layer, it cannot use relative position information which is needed for the copying mechanism, it has no semantic similarity needed for the moving mechanism). That said, the toy model of equation 2 is relevant in the sense that it can be used to **capture some of the dynamics underlying the learning of sparse attention and therefore of the induction head**. This is because the attention needed for the copying mechanism of the induction head is sparse (attention on the last token), and the one for the moving mechanism is sparse (attends to the same query token in the context).
>
> We hope this answers the questions of the reviewer and we remain available for additional clarification.

---

> ### Comment · Reviewer_AeaF · 2025-08-01
> **Thanks for your response**
>
> Thank you for your response and addressing my questions. I think this is a valuable contribution towards understanding training time emergence, and have maintained my initial rating of accept. I would encourage the authors to clarify the above points in their main paper as well, for reader convenience.

---

### Official Review · Reviewer_BBS6 · 2025-07-02

**Clarity:** 3
**Significance:** 3
**Originality:** 3
**Rating:** 5
**Confidence:** 4

**Summary:**

This paper investigates the emergence of “sparse attention” using a toy setup of training a small Transformer on a synthetic linear‐regression task. They show that the number of training steps to emergence scales with model size and sequence length, matching their theoretical derivation. They also find that repeating examples in the training data accelerates the emergence. Finally, they validate these findings on a more realistic associative‐recall task, confirming that sparse‐attention emergence governs performance there as well.

**Questions:**

Q: Do you observe any difference based on the position of the relevant token in the linear‐regression tasks? Would choosing later tokens help induce sparse attention more quickly?

Q: It seems intuitive that repeating information in the context aids retrieval. However, that doesn’t necessarily require sparse attention. I’m confused about how repetition causally drives attention to become sparse—how are these two phenomena related?

**Ethical Concerns:**

["NO or VERY MINOR ethics concerns only"]

**Final Justification:**

I remain my positive evaluation of this work.

**Limitations:**

Yes

**Quality:**

3

**Strengths And Weaknesses:**

Strengths

1. The paper provides a detailed theoretical analysis of sparse‐attention emergence in a toy linear‐regression setup and corresponding simplified Transformer model. It then extends this analysis to a more realistic associative‐recall task, showing that key observations generalize.

2. The authors skillfully bridge mechanistic interpretability and theory, moving from gradient‐based proofs to empirical validation.

3. This work offers a novel proof of concept for how mechanistic insights can inform realistic training‐data selection.

4. The experimental setup is clear and reproducible: the authors thoroughly explore different model sizes, sequence lengths, and optimization methods.

Weaknesses

1. Although the paper shows that data repetition accelerates sparse‐attention emergence, it only briefly discusses implications for tasks requiring diverse data. A deeper analysis of which tasks benefit most from repetition (e.g., those relying heavily on in‐context information) would strengthen the discussion.

2. Some observations—such as the proposed power‐law scaling—do not hold consistently across the linear‐regression and associative‐recall setups. Since both tasks remain somewhat artificial, it is unclear how these insights will generalize to real‐world applications.

---

> ### Author Rebuttal · Authors · 2025-07-30
>
> We thank the reviewer for their positive feedback and thoughtful review. We address the weaknesses they raised and the questions they asked below:
> 1. **Weakness: no discussion of which task could benefit from these effects.** We thank the reviewer for pointing out this blind spot in our argumentation. We believe **our findings to be general and not limited to tasks that rely on heavy in-context manipulations**. We believe that **some sparse context selection is needed** in order to see the emerging behavior we characterize, which might be relatively common given the sparsity of trained LLMs.
> We expect **cross-sample repetition to be particularly helpful in tasks in which learning the feedforward mapping is hard** (high $d$ in our analysis, which this form of repetition effectively reduces). This includes learning facts – the findings of Zucchet et al. 2025 are one example of the benefits of cross-sample repetition – but may extend to some other tasks like reasoning-heavy tasks.
> **In-context repetition mostly when learning from very long sequences** (as they decrease the effective sequence length). It is a natural feature of language (it helps humans to process texts), and we expect that reducing it would actually considerably slow down the learning of language models.
> We have included more detail about these considerations in the discussion and we hope that our work will motivate future studies to investigate these questions.
> 2. **Weakness: theoretical power laws don’t exactly extend to other tasks.** Indeed, each task seems to have its own power law and its specifics are task-, architecture- and optimizer- dependent (our ablations of Appendix B.3 shows quite strong power law variations already on the single-location linear regression task only). We believe that the fact that there exists such a power law for more sophisticated tasks (see also Figure 2 right of Zucchet et al. 2025) is a good sign that **the core phenomenon, though not the exact coefficients, is robust and may generalize to real-world applications**.
> 3. **Question: does the position of the selected token affect learning speed?** **We have no reasons to believe that it has a strong impact.** When looking at Figure 11, which shows learning dynamics for different seeds (and thus different locations), all of them have similar dynamics.
> 4. **Question: role of in-context repetition and does repetition causally drive sparse attention?** The reviewer is right that the effect of in-context repetition is rather intuitive: if the same information is repeated multiple times in the context, it becomes more correlated with the token to predict and thus learning is faster. Our theory captures that analytically. **Repetition affects the learning of sparse attention** (as per the argument above), but **it does not causally drive sparse attention** in the sense that sparse attention would arise even without repetition (because it is useful to solve the task).
>
> We hope this clarifies our results, and we remain available during the rebuttal period for additional clarifications.

---

> > ### Comment · Reviewer_BBS6 · 2025-08-03
> >
> > Thank you for the responses to my questions.

---

### Decision · Program_Chairs · 2025-09-17

**Decision:**

Accept (oral)

**Comment:**

After careful consideration of the paper and the reviewer comments, I recommend accepting this paper. The paper provides a novel, mechanistic account of how sparse attention patterns emerge during training, demonstrating through both a simplified theoretical model and empirical validation that this emergence follows power-law dynamics influenced by task structure, architecture, and optimizer choice, and is significantly accelerated by data repetition. While the theoretical setup (single-location linear regression) is indeed a simple model, it does the job of providing a minimal example of sparsification. The authors' thorough and effective responses, which satisfied or partially satisfied all reviewers, including the initially skeptical oc9H, will significantly strengthen the manuscript if applied, which I strongly recommend the authors to do.